# Nonlinear exposure-response associations of daytime, nighttime, and day-night compound heatwaves with mortality amid climate change

Jiangdong Liu [1], Ho Kim [2], Masahiro Hashizume [3], Whanhee Lee[4], Yasushi Honda[5,6], Satbyul Estella Kim [5,6], Cheng He [7], Haidong Kan [1,8] ✉ & Renjie Chen [1] ✉

Heatwaves are commonly simplified as binary variables in epidemiological studies, limiting the understanding of heatwave-mortality associations. Here we conduct a multi-country study across 28 East Asian cities that employed the Cumulative Excess Heatwave Index (CEHWI), which represents excess heat accumulation during heatwaves, to explore the potentially nonlinear associations of daytime-only, nighttime-only, and day-night compound heatwaves with mortality from 1981 to 2010. Populations exhibited high adaptability to daytime-only and nighttime-only heatwaves, with non-accidental mortality risks increasing only at higher CEHWI levels (75th–90th percentiles). In contrast, compound heatwaves posed a super-linear increase in mortality risks after the 25th percentile of CEHWI. Associations of heatwaves with cardiovascular mortality mirrored those with non-accidental mortality but were more pronounced at higher CEHWI levels, while significant associations with respiratory mortality emerged at low-to-moderate CEHWI levels. These results highlight the necessity of considering the nonlinear health responses to heatwaves of different types in disease burden assessments and heatwave-health warning systems amid climate change.

Heatwaves, characterized as extended periods of extreme heat[1], constitute a unique form of extreme temperature events with wide-ranging and profound health implications[2–7]. A multitude of epidemiological studies has documented associations between heatwaves and elevated risks of mortality from a diverse range of diseases[8–14]. Furthermore, evidence suggests that populations could exhibit adaptive physiological responses to elevated temperatures[15–19], such as vasodilation, increased plasma volume, and enhanced sweat rates. These short-term, albeit bounded, physiological adaptations may lead to nonlinear responses to heatwaves with varying cumulative heat intensities or distinct heat exposure patterns. Given these adaptive capacities and the knowledge that the health effects of high

[1]School of Public Health, Shanghai Institute of Infectious Disease and Biosecurity, Key Lab of Public Health Safety of the Ministry of Education and NHC Key Lab of Health Technology Assessment, Fudan University, Shanghai, China. [2]Department of Biostatistics and Epidemiology, Graduate School of Public Health, Seoul National University, Seoul, Republic of Korea. [3]Department of Global Health Policy, Graduate School of Medicine, The University of Tokyo, Tokyo, Japan. [4]School of Biomedical Convergence Engineering, Pusan National University, Yangsan, South Korea. [5]Center for Climate Change Adaptation, National Institute for Environmental Studies, Tsukuba, Japan. [6]Faculty of Health and Sport Sciences, University of Tsukuba, Tsukuba, Japan. [7]Institute of Epidemiology, Helmholtz Zentrum München–German Research Center for Environmental Health, Neuherberg, Germany. [8]Children's Hospital of Fudan University, National Center for Children's Health, Shanghai, China. ✉e-mail: kanh@fudan.edu.cn; chenrenjie@fudan.edu.cn

temperatures are not strictly linear in prior studies[20–23], we hypothesize that the relationship between heat anomalies during heatwaves and health outcomes is not necessarily linear. However, unlike non-optimal temperature treated as a continuous variable[20–23], prevailing risk assessment studies consistently oversimplified heatwaves as a binary variable[8–14], which fails to capture complex associations of heatwaves with mortality. Understanding variations in the health impacts of heatwaves across different types and cumulative heat scales could inform the development of adaptive strategies, such as redesigning more targeted and effective heatwave-warning systems amid climate change[24–26].

Recognizing the limitations of the prevailing binary definition, our study calculated the Cumulative Excess Heatwave Index (CEHWI), which measures cumulative excess heat during heatwaves relative to a predefined threshold (detailed calculation described in Methods). This index captures both the fluctuations in heat magnitude and the continuous nature of cumulative excess heat during heatwaves, thus providing a more refined lens on the complex and potentially nonlinear relationships between heatwaves and mortality.

In the context of global warming, the patterns of heatwave occurrence are shifting from traditional daytime-only heatwaves (DHW) to nighttime-only (NHW) and day-night compound heatwaves (CHW)[27–29]. Recent studies have highlighted the more pronounced adverse effects of nighttime and compound heat compared to daytime heat on health outcomes[13,14,30]. Consequently, focusing solely on traditional DHW risks may underestimate the broader health impacts of heatwaves in a changing climate. While existing studies have compared the health effects of these three heatwave types[13,31,32], they have been limited to single countries. To date, few studies have compared these health effects across multiple countries.

In this multi-country study, we establish exposure-response curves for the associations of the CEHWI with daily non-accidental, cardiovascular, and respiratory mortality across East Asia for three distinct heatwave types. We further quantify the mortality burden associated with these heatwaves based on the exposure-response curves. We observe that populations exhibit greater adaptability to DHW and NHW, with nonlinear health effects, as their exposure-response curves only significantly increase after approximately the 90th percentile of the CEHWI. In contrast, populations show higher sensitivity to CHW, which pose significantly greater disease risks and burdens, particularly for cardiovascular diseases, across East Asia. These findings challenge the suitability of using a simple binary variable to define heatwaves in health risk assessments, suggesting the need to capture the complex, nonlinear health effects of heatwaves under global warming.

## Results

### Descriptive statistics
Our dataset includes daily non-accidental, cardiovascular, and respiratory deaths across 28 cities in East Asia during the summer seasons from 1981 to 2010 (Supplementary Table 1). A total of 1,464,632 non-accidental deaths were recorded, with daily deaths ranging from 7 in Tianjin to 142 in Tokyo. Cardiovascular diseases (CVDs) constituted approximately 33.8% of non-accidental deaths, while respiratory diseases (RDs) accounted for about 10.8%. The average frequencies of DHW, NHW, and CHW were 3.3, 3.2, and 3.0 days per summer, respectively (Supplementary Table 1). DHW and NHW were most frequent in South Korea (4.3 and 3.9 days, respectively), whereas CHW was most common in China (3.7 days).

### Relationships between three types of heatwaves and mortality
The pooled exposure-response curves for CEHWI and non-accidental mortality in East Asia exhibit nonlinearity for DHW and NHW but are nearly linear for CHW, with considerable variations across heatwave types (Fig. 1). Specifically, for DHW and NHW, the risk of non-

accidental mortality was modest or insignificant below moderate levels of CEHWI but surged beyond the 75th–90th percentiles of CEHWI. In contrast, for CHW, the associated mortality risk consistently increased almost across the full CEHWI range without discernible thresholds. Across countries, the exposure-response curves were generally consistent, except for DHW in China, where the impact emerged at lower CEHWI levels and continued to rise.

For CVD mortality, the exposure-response curves generally resembled those for non-accidental mortality in their nonlinear trend, but the slope was steeper at high CEHWI levels (above the 90th percentile), particularly for CHW (Fig. 2). In contrast, for RD, the exposure-response curves also displayed nonlinear trends, but mortality risks emerged at lower CEHWI levels for DHW and CHW (Fig. 3). However, an exception was observed in China, where the impact of CHW on RD became significant only when CEHWI exceeded the 75th percentile.

The lag structures for the associations of extreme DHW, NHW, and CHW (the 90th percentile of CEHWI distribution) with non-accidental, cardiovascular, and respiratory mortality in East Asia were largely similar (Fig. 4). Mortality risks associated with heatwaves were observed on the day of exposure and persisted for 2–5 days, contingent upon the heatwave type and cause of mortality. CHW-related mortality risks were more prolonged and significant compared to DHW and NHW, with the lower bound of the confidence interval (CI) remaining significant for 3–5 days. Risks associated with NHW for non-accidental and cardiovascular mortality also lasted about five days but were only marginally significant during these days. For DHW, the effects were significant for only 2–3 days across mortality causes. There were no significant differences in lag patterns for the same type of heatwaves across countries (Supplementary Fig. 1), except for DHW in China, where the lag pattern was longer than in other countries.

### Mortality risk and burden
The pooled mortality risks varied across the three heatwave types (Fig. 5), with CHW associated with significantly higher risks than DHW and NHW. However, the mortality risks related to DHW did not significantly differ from those due to NHW across all disease causes. For instance, regarding non-accidental mortality, the relative risks (RRs) associated with the 90th percentile of CEHWI were 1.04 (95% CI: 1.00, 1.08) for DHW, 1.03 (1.01, 1.04) for NHW, and 1.18 (1.15, 1.21) for CHW. For cardiovascular and respiratory mortality, the RRs associated with CHW were 1.29 (1.24, 1.35) and 1.13 (1.07, 1.20), respectively.

The pooled mortality burden linked to CHW was consistently higher than that due to DHW and NHW (Fig. 5). For non-accidental and cardiovascular mortality, the attributable fractions (AFs) associated with CHW largely surpassed those for DHW and NHW. Specifically, the AFs of CVD mortality related to DHW, NHW, and CHW were 0.14% [95% empirical confidence intervals (eCIs): 0.06%, 0.21%], 0.04% (0.01%, 0.05%), and 0.47% (0.36%, 0.52%), respectively. Moreover, the DHW-related burden was insignificantly larger than that attributable to NHW for non-accidental, cardiovascular, and respiratory mortality. The specific national mortality risks and burdens showed no significant differences from the pooled trends (Supplementary Table 2).

### Results of effect modification
There was significant heterogeneity in the pooled relationships between the three types of heatwaves and non-accidental mortality, with $I^2$ statistics of 23.4% for DHW, 24.6% for NHW, and 64.1% for CHW in meta-regression models that did not include any meta-predictors (Supplementary Table 3). For DHW, population size was the only significant meta-predictor, reducing $I^2$ statistics to 17.0% in the univariable meta-regression. Regions with larger populations, defined as cities with more than 3.3 million residents (the median population size across all cities in this study), exhibited an elevated

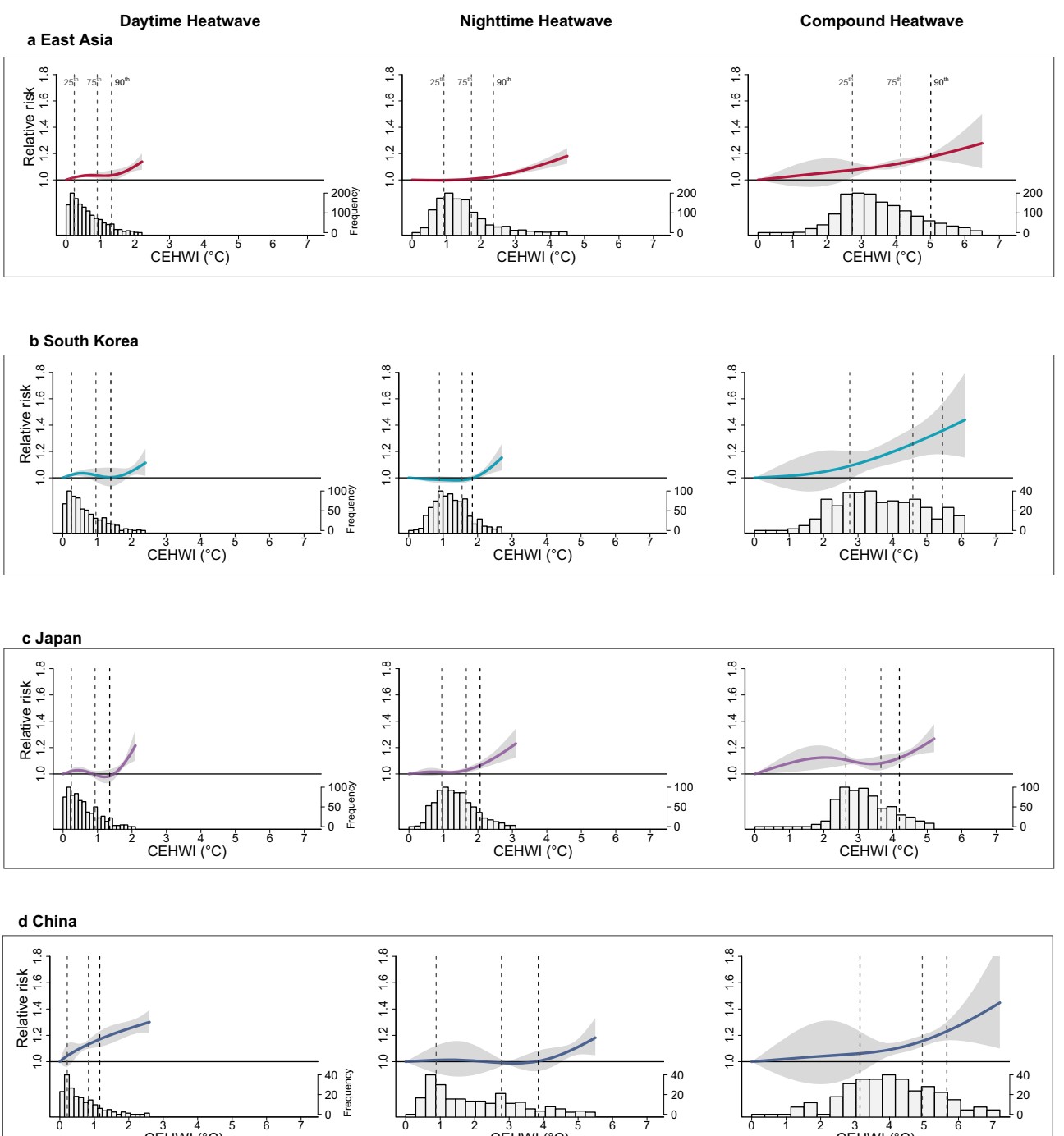

**Fig. 1 | Pooled exposure-response relationship curves of the cumulative excess heatwave index (CEHWI) for daytime, nighttime, and day-night compound heatwaves in relation to non-accidental mortality, along with the corresponding CEHWI distributions. a** East Asia; **b**, South Korea; **c**, Japan; **d**, China. The solid lines represent the estimated cumulative relative risks (point estimates) of risk associated with DHW (Supplementary Fig. 2). However, no significant meta-predictors were identified for NHW and CHW in either the univariable or multivariable meta-regression models, and $I^2$ statistics did not decrease noticeably when including meta-predictors (Supplementary Table 3).

mortality on heatwave days compared to non-heatwave days, while the shaded areas denote the corresponding 95% confidence intervals. The vertical dotted lines, from left to right, indicate the 25th, 75th, and 90th percentiles of the regional CEHWI distribution during heatwaves. Source data are provided as a Source Data file.

### Results of sensitivity analyses

The pooled exposure-response curves for associations between the three types of heatwaves and mortality remained largely unchanged after controlling for concentrations of particulate matter with an aerodynamic diameter ≤10 μm ($PM_{10}$) and ozone ($O_3$) (Supplementary Figs. 3–5), suggesting that these pollutants are not significant confounders. For different model parameter settings, the AFs remained stable after modifying nonlinear functions or knots in cross-basis functions in the first-stage analysis (Supplementary Table 4: tests 1–3). Changes in the degrees of freedom (*dfs*) for the nonlinear function of relative humidity did not substantially alter the results (Supplementary Table 4: tests

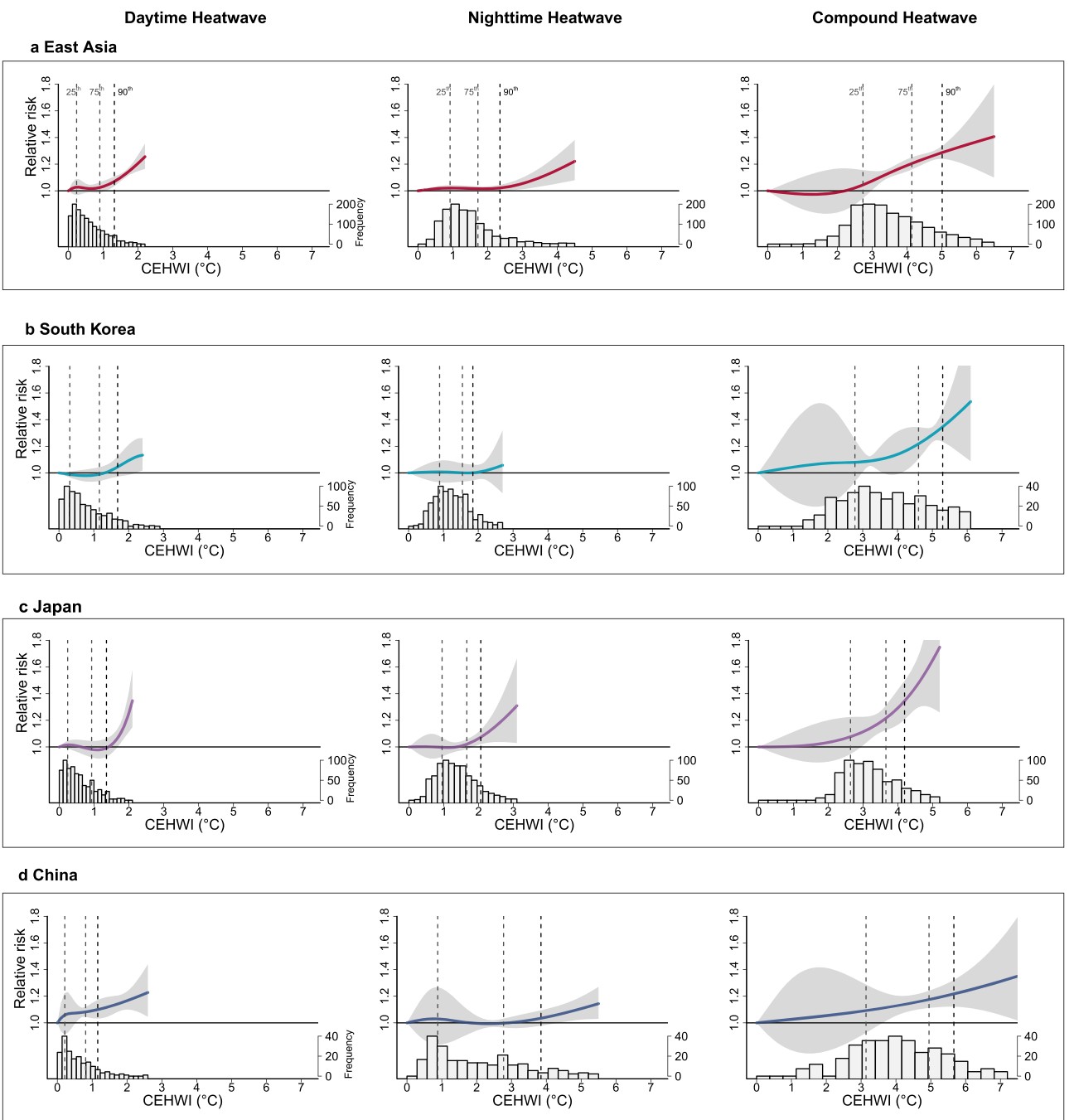

**Fig. 2 | Pooled exposure-response relationship curves of the cumulative excess heatwave index (CEHWI) for daytime, nighttime, and day-night compound heatwaves in relation with cardiovascular disease mortality, along with the corresponding CEHWI distributions. a** East Asia; **b** South Korea; **c** Japan; **d** China. The solid lines represent the estimated cumulative relative risks (point estimates) of mortality on heatwave days compared to non-heatwave days, while the shaded areas denote the corresponding 95% confidence intervals. The vertical dotted lines, from left to right, indicate the 25th, 75th, and 90th percentiles of the regional CEHWI distribution during heatwaves. Source data are provided as a Source Data file.

4–5). The 90th percentile of the summer temperature distribution significantly exceeds the minimum mortality temperature (MMT) (Supplementary Fig. 6), supporting its suitability as a threshold for defining extreme heat events. With increasing thresholds and durations in the heatwave definition, mortality risks of heatwaves showed an upward trend. Nonetheless, the risks related to CHW were consistently higher than those due to DHW and NHW (Supplementary Table 5). For the binary form of heatwaves, the effect of CHW remained higher than that of DHW and NHW (Supplementary Table 5).

## Discussion

By employing the CEHWI, this multi-country study across 28 East Asian cities offers a refined approach to capturing the complex relationships between short-term heatwave exposure and mortality risks, moving beyond the conventional binary definition of heatwaves in epidemiological studies and health risk assessments. Additionally, this study compares the health effects of DHW, NHW, and CHW from a multi-country perspective. Our findings uncover intricate and distinct exposure-response patterns between different heatwave types and cause-specific mortality, warranting a departure from conventional

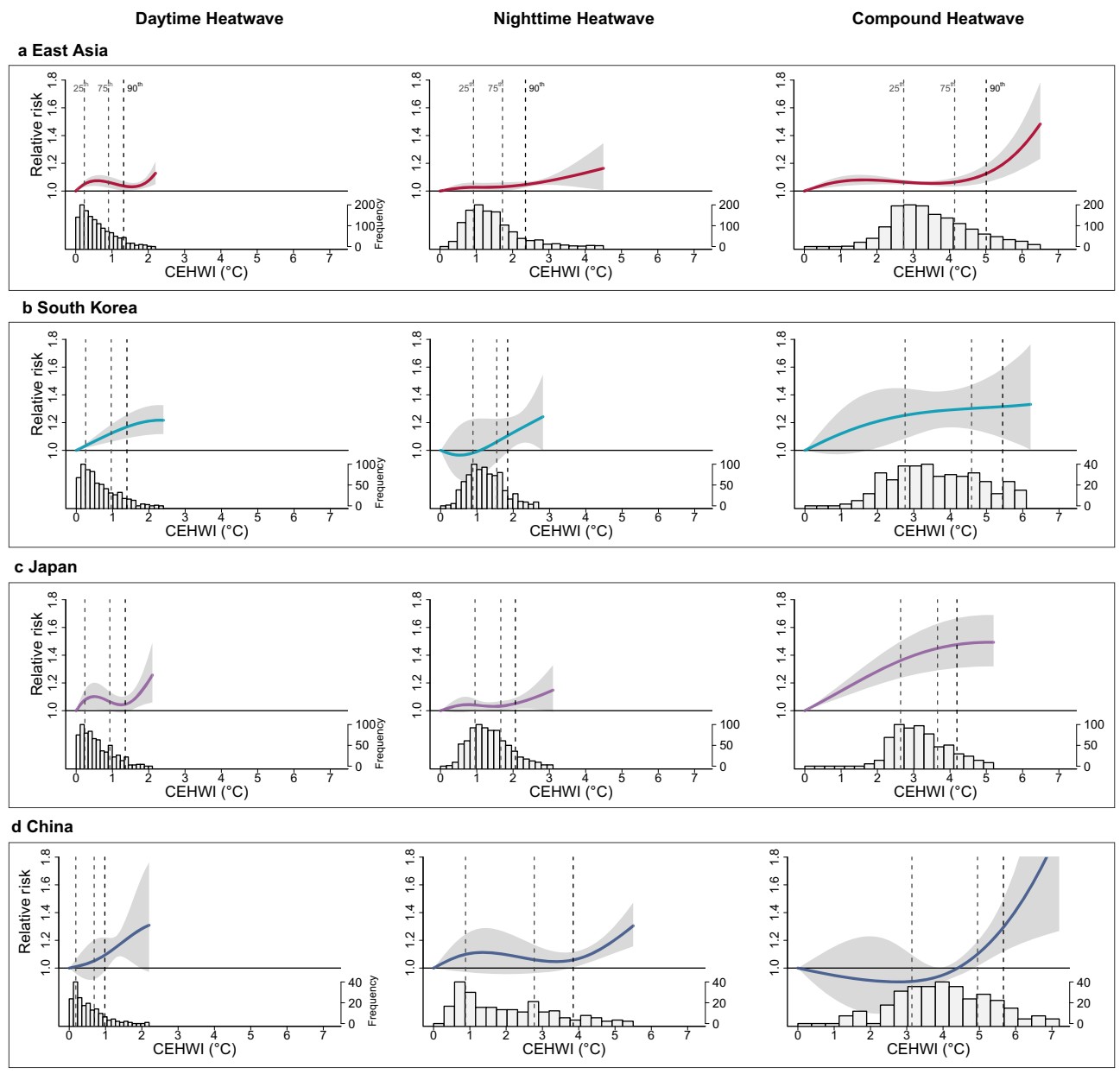

**Fig. 3 | Pooled exposure-response relationship curves of the cumulative excess heatwave index (CEHWI) for daytime, nighttime, and day-night compound heatwaves in relation with respiratory disease mortality, along with the corresponding CEHWI distributions. a** East Asia; **b** South Korea; **c** Japan; **d** China. The solid lines represent the estimated cumulative relative risks (point estimates) of mortality on heatwave days compared to non-heatwave days, while the shaded areas denote the corresponding 95% confidence intervals. The vertical dotted lines, from left to right, indicate the 25th, 75th, and 90th percentiles of the regional CEHWI distribution during heatwaves. Source data are provided as a Source Data file.

risk assessments that focused solely on traditional DHW. Notably, our study reveals the significantly higher mortality risk and burden from CHW compared to DHW and NHW. Overall, this study expands health risk assessment for heatwaves in the context of global warming, providing insights for the development of heatwave-related public health interventions.

One of the contributions of this study lies in the exploration of the potential nonlinear relationship between heatwaves and mortality. Unlike binary definitions, incorporating cumulative excess heat during heatwaves, which accounts for continuous heat exceeding specific thresholds, provides a more accurate reflection of their hazardous impacts. This is because these threshold-exceeding temperatures typically surpass the limits of human adaptability[33,34]. Importantly, predicated on this metric, our findings emphasize the nonlinear nature of the impacts of excess heat on mortality risk

during heatwaves, potentially attributed to the short-term adaptability of populations[16,19,25]. Additionally, our estimated disease burden is relatively lower compared to those from studies using a binary approach[13,30]. For example, two studies in Chinese cities reported attributable fractions (AFs) of 0.96% and 1.23% for compound heat-related non-accidental mortality, respectively, both higher than our estimate (0.31%). This discrepancy likely arises from our use of exposure-response curves to estimate the burden, which considers population adaptation to lower-to-moderate cumulative heat intensities. In contrast, binary definitions oversimplify this complex response, potentially overestimating the burden from lower-intensity heatwaves. Furthermore, our study unveiled an additional layer of complexity in population adaptability, manifesting as a discernible adaptability threshold that varies across heatwave types, diseases, and countries. This variability is likely attributed to

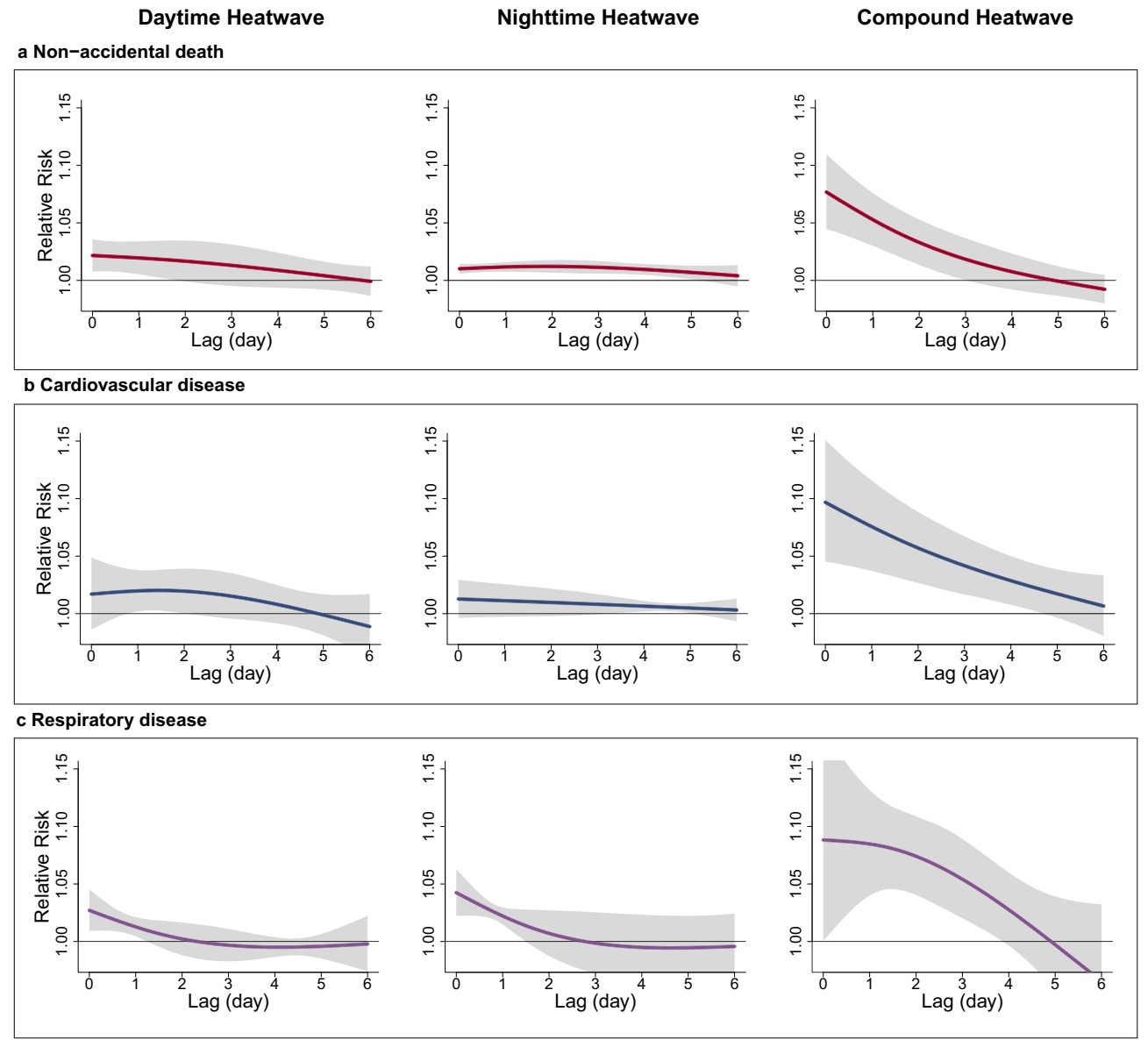

**Fig. 4 | Lag structures for the associations of daytime, nighttime, and day-night compound heatwaves with non-accidental, cardiovascular disease, and respiratory disease mortality in East Asia. a** Non-accidental death; **b** Cardiovascular disease; **c** Respiratory disease. The solid lines represent the estimated relative risks (point estimates) of mortality for each lag day associated with specific heatwave types compared to non-heatwave days, while the shaded areas indicate the corresponding 95% confidence intervals. Source data are provided as a Source Data file.

differences in health effects of distinct heatwave exposure patterns, the mechanisms by which different diseases respond to heatwaves, and variations in regional geography, climate, and socioeconomic conditions. Consequently, future epidemiological studies and health risk assessments should avoid simplistic binary representations of heatwaves and further consider the nonlinear health effects of heatwaves.

Previous studies have projected that the frequency of CHW would gradually exceed that of DHW and NHW due to climate change[27,28], with CHW exhibiting larger health effects than the other two types[13,14]. Our study further underscores the distinct response of populations to these different types of heatwaves. For the population's adaptability to DHW with low to medium CEHWI (below the 90th percentile), this may be explained by both short-term adaptation and long-standing acclimatization to the conventional hot-day and cool-night pattern[19,35,36]. Similarly, NHW causes significant impact only at medium-to-high CEHWI (above approximately the 80th percentile), likely due to

nocturnal protection offered by building features (e.g., cool and evaporative roofs[15]) and indoor facilities (e.g., air conditioning systems[37]). However, without information on individuals' daytime working environments (e.g., work indoors or outdoors), we acknowledge that the observed DHW adaptation may also be influenced by the effectiveness of buildings and air conditioning. Future studies should explore the health mechanisms underlying daytime and nighttime heat exposure at the individual level to better elucidate their differing impacts. Notably, the population did not exhibit apparent adaptability to CHW, which manifested a continuously increasing excess risk without an apparent threshold. This is possibly attributable to a double-strike mechanism[14,38], wherein NHW impedes physiological processes, including the sleep-wake cycle and thermoregulation, thereby amplifying the impacts of subsequent DHW. Our study indicates that mortality risks associated with CHW were more pronounced compared to those related to DHW and NHW on a multicountry scale, which aligns with previous studies conducted in single countries[14,30,38].

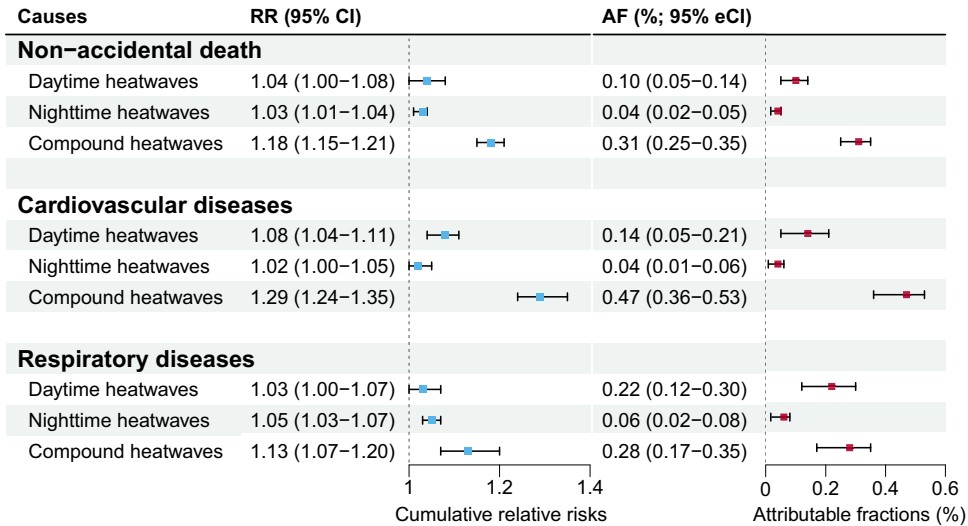

**Fig. 5 | Cumulative relative risks (RRs) and attributable fractions (AFs) of non-accidental, cardiovascular disease, and respiratory disease mortality associated with daytime, nighttime, and day-night compound heatwaves in East Asia.** RRs and AFs are depicted as squares (centres of the error bars), with the error bars representing the 95% confidence intervals. RRs were defined as the risks at the mean of the 90th percentile of cumulative excess heatwave index distributions compared with the risks on non-heatwave days. Source data are provided as a Source Data file. Abbreviation: eCIs empirical confidence intervals.

Similar to previous studies[9,13,30], our findings reveal differing risks and burdens associated with heatwaves across CVD and RD mortality. Our estimated exposure-response curves indicate that these differences may arise from the distinct response mechanisms of cardiovascular and respiratory diseases to heatwaves. For CVD, the exposure-response curves closely align with non-accidental mortality at low-to-medium CEHWI levels but become more pronounced at high CEHWI levels, although with regional differences. This may stem from the physiological impacts of heatwaves on the cardiovascular system, primarily involving the failure of compensatory mechanisms[39,40]. These failures typically occur at higher heat intensities and are marked by dehydration[41], electrolyte imbalances[42], and increased blood viscosity[40]. In contrast, RD showed a rapid response to heatwaves at lower CEHWI levels, with significant impacts emerging at lower CEHWI levels, particularly for DHW or CHW. Specifically, respiratory conditions could be exacerbated due to increased respiratory rates and fluid loss[43], which can occur at lower heat levels, leading to mucus thickening and airway inflammation[44]. For example, in South Korea, DHW was associated with significant risks for RD at low CEHWI levels, whereas risks for CVD mortality became significant only after the CEHWI exceeded approximately the 90th percentile. These distinct physiological reactions emphasize the need for tailored disease-specific heatwave warning systems. However, the impacts of DHW and CHW on RD in China showed an insignificant trend at low-to-moderate CEHWI levels, in contrast to CVD, where effects emerged at similar CEHWI levels. This discrepancy may be due to the fact that individuals at high risk for RD might have already been severely affected by CVD, which tends to respond to lower-intensity heatwaves in China. As a result, the impact of heatwaves on RD only became significant when heat intensities were high enough to affect those who could initially withstand lower-intensity heatwaves. Further research is needed to explore the physiological mechanisms driving these disease-specific responses to heatwaves.

Identifying potential effect modifications for the heatwave-related mortality risk concerning climatic and socioeconomic characteristics is crucial for developing evidence-based region-specific health protection plans. Our findings reveal certain heterogeneity in the effects of DHW, with population size emerging as a significant contributing factor, which is consistent with previous researches[45,46]. Cities with larger populations (i.e., those with more than 3.3 million

people, the median population size across all cities in this study) experience increased heatwave-related mortality risks. This is likely attributable to more intense and frequent exposure in densely populated urban areas, where the urban heat island effect is more pronounced[45,47]. This might also explain why the effects of DHW in China exhibited longer lags and became significant for non-accidental mortality at lower CEHWI levels, as our Chinese dataset included some of the most populous cities in East Asia, such as Shanghai, Beijing, and Guangzhou. Furthermore, China's relatively limited public infrastructure capacity, specifically for heat resilience, compared to Japan and South Korea, may amplify the adverse impacts of DHW. The heterogeneity in NHW effects aligns closely with that of DHW, while CHW effects exhibited greater heterogeneity surpassing both DHW and NHW. This heightened heterogeneity in CHW effects can be attributed to its nature as a prolonged event spanning both daytime and nighttime. CHW not only encompasses the individual impacts of DHW and NHW but also introduces interactive effects between daytime and nighttime exposures, contributing to increased complexity. However, this study did not observe significant effect modifiers for NHW and CHW, which warrants further exploration in future studies.

Our findings hold several policy implications. Firstly, the traditional binary approach in epidemiological studies may oversimplify the nonlinear relationship between heatwaves and population health, overestimating the heatwave-related burdens. A more comprehensive assessment, accounting for these nonlinearities, can serve as a scientific guide for policymakers in optimizing risk communications and resource allocation, particularly in the context of climate change. Secondly, the design of heatwave warning systems should incorporate different heatwave types, with a heightened focus on CHW, which is expected to cause larger health burdens. Moreover, attention should also be directed towards high-intensity heatwaves, regardless of their specific types. Thirdly, strategic investments in public facilities, such as green spaces, and individual-level protections like air conditioning, are paramount in densely populated regions to mitigate the mortality risk associated with heatwaves. Fourthly, given the identified varying thresholds in health effects for different heatwave types, policymakers are encouraged to adopt more effective interventions to increase the adaptive capacity of populations, thus increasing the population's health adaptation thresholds to 90th percentile CEHWI or above, which could

significantly mitigate the heatwave-related burdens. These interventions should target multiple levels, including individual, community, institutional, and public policy domains[11]. Lastly, our results underscore the urgency of robust mitigation strategies to counteract the escalating threat of global warming, which is anticipated to lead to more frequent compound events such as CHW.

Limitations should also be acknowledged. Firstly, despite utilizing temperature data from the well-established reanalysis dataset (ERA5-Land) with a spatial resolution of $0.1° \times 0.1°$ and hourly temporal resolution, the current study was subject to inherent exposure errors due to the unavailability of personal temperature data. However, these errors are likely random and are considered to underestimate the true effects[48]. Secondly, our study focused on 28 cities across three East Asian countries; consequently, caution should be exercised when extrapolating our findings to the entire East Asian region or other geographical areas. Thirdly, the study periods varied among the selected cities. The relatively shorter study periods for Chinese cities may attenuate the statistical power for risk estimation in these cities. Nevertheless, this is not expected to significantly impact the aggregated results, given the larger number of Chinese cities included in the overall analysis.

In conclusion, this multi-country analysis provides a detailed examination of the complex associations between heatwaves and mortality, emphasizing the nonlinear exposure-response patterns across three types of heatwaves. While DHW and NHW showed insignificant effects at low-to-moderate CEHWI levels, CHW displayed a super-linear increase in mortality risk, underscoring the critical need to enhance population adaptation to CHW. The distinct responses of CVD and RD to heatwaves further highlight the necessity of incorporating disease-specific considerations in heatwave-warning systems. These findings highlight the importance of accounting for different heatwave types and their cumulative intensities to effectively mitigate the growing heatwave-related health burden in a warming world.

## Methods
### Historical mortality and environmental data
Our study incorporated mortality data from 28 cities across three East Asian countries, namely South Korea (six cities), Japan (seven cities), and China (15 cities), covering the period from 1981 to 2010 (Supplementary Fig. 7). The data collection periods varied within this timeframe across locations (Supplementary Table 1). These cities encompass diverse climates, from tropical to temperate and coastal to inland zones, as well as a range of socioeconomic conditions, spanning middle- to high-income levels in East Asia.

The mortality data used in our analysis were collected, and verified by government agencies in each country or region, ensuring their reliability and validity[49,50]. We obtained daily death records from the Korea National Statistics Office in South Korea, the Ministry of Health, Labor, and Welfare in Japan, the Census and Statistics Department in Hong Kong, and the local Centers for Disease Control and Prevention in mainland China. Utilizing the International Classification of Diseases Tenth Revision (ICD-10), we extracted data on non-accidental causes (coded as A00–R99), CVD (coded as I00–I99), and RD (coded as J00–J99). These three mortality outcomes were selected due to their sensitivity to heat exposures and their high public health relevance[2,51], which ensures a comprehensive evaluation of the mortality risks associated with heatwaves by capturing both overall and specific impacts on major diseases. To focus on heatwave-related mortality risk and burden, we confined our study period to the summer season (June to August), in line with established research norms[14,27,28,30].

Due to the absence of hourly temperature observations for all selected cities, we used the land component of the fifth generation of European Reanalysis (ERA5-Land) product for temperature data, which provides $0.1° \times 0.1°$ spatial and hourly temporal resolutions. We obtained relative humidity data from the Korean Meteorological Office, the Japanese Meteorological Agency, and the Chinese Meteorological Data Sharing Service System. To adjust for air pollution, we acquired daily concentrations of $PM_{10}$ and $O_3$ from the Korean Research Institute of Public Health, the Japanese National Institute for Environmental Studies, and the China National Environmental Monitoring Centre.

### Definitions of heatwave and CEHWI
A heatwave is conventionally defined as an event where daily temperatures surpass a certain absolute or relative threshold for a continuous period[1,9,10,24]. Given climatological disparities across East Asia, we applied relative thresholds to identify local temperature extremes and enable cross-regional comparisons. A day or night is classified as 'heat' when the daily maximum (Tmax) or minimum (Tmin) temperature exceeds the 90th percentile of its long-term distribution (1981–2010)[14,28,30]. Instead of using a seasonal-based percentile, a daily-based percentile calculated from a 15-day sliding window centered on each calendar day over 1981–2010 was employed. This approach allows for the identification of heatwaves occurring in early or late summer and distinguishes between intense and moderate events during the peak of summer[28,52].

Since the physical conditions triggering heatwaves during the day and night are distinct[27,29,52], the corresponding physiological responses to heat stress may also differ. For example, daytime heatwaves are primarily caused by solar radiation and anticyclonic conditions leading to surface heating[28,53]. In contrast, nighttime heatwaves result from insufficient nocturnal radiative cooling of accumulated daytime heat[54], with dry soil conditions and strong day-night land-air interactions further intensifying compound heatwaves[55]. Based on these differences, we classified heatwaves into three distinct types: (a) DHW, which requires at least two consecutive hot days without preceding hot nights (Tmax ≥ 90th percentile and Tmin < 90th percentile); (b) NHW, which necessitates a minimum of two consecutive hot nights without following hot days (Tmin ≥ 90th percentile and Tmax < 90th percentile); and (c) CHW, which is characterized by at least two consecutive days with both hot days and nights (Tmax ≥ 90th percentile and Tmin ≥ 90th percentile). The use of the 90th percentile and a two-day duration achieves a careful balance, taking into account both the extremity of the event and the sample size for analysis[14]. To synchronize the categorization of heatwaves (exposure) with their subsequent mortality consequences, each day was subdivided into three distinct periods: nighttime (midnight–sunrise), daytime (sunrise–sunset), and evening (sunset–midnight). DHWs manifest during the daytime, NHWs occur during the nighttime and evening, and CHWs encompass both types.

We employed the CEHWI to represent the cumulative and dynamic nature of heatwave intensities, which was modified according to a cumulative heat indicator developed in a prior study[33]. This index better captures the accumulation and persistence of heat at different levels during heatwaves, avoiding simplifying heatwaves as binary variables. Additionally, it specifically focuses on the additional heat that exceeds heatwave thresholds, as it is the excess temperature beyond these thresholds that poses the greatest risk to the population health[33,34]. This approach serves as a nuanced measure to explore the non-linear health impacts of heatwaves, enabling a more accurate assessment of their hazardous effects. This index was computed as the sum of the hourly temperature difference between each heatwave within the respective periods and the calendar-day heatwave threshold, as shown in Eqs. (1) to (3):

$$CEHWI_{ig} = \sum_{j}^{N_{ig}} (t_{jg} - Thr_{ig}) \times I_{Thr_{ig}}(t_{jg}) / n_{ig} \tag{1}$$

$$Thr_{ig} = \begin{cases} T_{\max g} & i=1 \\ T_{\min g} & i=2 \\ T_{\text{mean} g} & i=3 \end{cases} \qquad (2)$$

$$I_{Thr_{ig}}(t_{jg}) = \begin{cases} 0 & t_{jg} \leq Thr_{ig} \\ 1 & t_{jg} > Thr_{ig} \end{cases} \qquad (3)$$

where $i$ refers to the type of heatwaves, $N_{ig}$ represents the total hours of daytime ($i=1$) or nighttime and/or evening ($i=2$) or whole day ($i=3$) in day $g$. $t_{jg}$ is the temperature at hour $j$ in day $g$, $Thr$ is the local daily-based temperature threshold, and $I_{Thr}$ is the indicator variable. $n_{ig}$ represents the total hours during heatwave periods in which $I_{Thr}$ is equal to 1.

## Statistical analysis

Our study utilized a well-established two-stage analytical framework to comprehensively estimate the complex relationship between CEHWI and daily mortality across the study period in each location[15,51].

In the first stage, we applied a generalized additive model with a quasi-Poisson link function, coupled with a distributed lag non-linear model to uncover the delayed and potentially non-linear effects of CEHWI on each cause of mortality for each city[56]. The cross-basis function of CEHWI employed a natural cubic spline with two internal knots positioned at the 50th and 90th percentiles of city-specific CEHWI distributions[57]. Additionally, a natural cubic spline over a lag of one week (0–6 days)[8] with three *dfs* was incorporated. To control for potential confounders, we incorporated a natural cubic spline with three *dfs* for relative humidity, a natural cubic spline with four *dfs* for day of the season, an indicator variable for year to account for long-term trends, and a day-of-the-week indicator to control for weekly patterns. These modeling decisions, which align with previous studies[51,58–60], underwent scrutiny during sensitivity analysis to ensure robustness.

In the subsequent analysis, we employed a multivariate meta-regression approach to pool the city-specific cumulative associations between CEHWI and mortality over a week[51,61], which was conducted at both national (South Korea, Japan, and China) and regional (East Asia) levels. The dependent variables in the models were the three sets of reduced spline coefficients, representing the cumulative association curves across lags for each city, and the coefficient estimates were obtained using the restricted maximum likelihood estimator. The city-specific spline of CEHWI was set relative to normal days (CEHWI = 0) for each city, which served as the baseline to calculate RRs for different CEHWI levels. The exposure-response curves were delineated with boundary knots set at the averages of the 99th percentile of the city-specific CEHWI distribution. This second-stage analysis also illustrated lag patterns in mortality risks associated with the average 90th percentile of the CEHWI distribution (defined as extreme heat in heatwaves).

## Effect modification analysis

Considering the potential spatial heterogeneity, we employed uni-variable meta-regression models incorporating the following city-specific attributes separately: Koppen–Geiger climate classification, gross domestic product (GDP) per capita, summer average temperature, summer temperature range, population size, latitude, and longitude. The statistical significance of the meta-predictors in accounting for heterogeneity was assessed using the Wald test. We also fitted a multivariable meta-regression model including all predictors except for latitude and longitude to avoid multicollinearity. Finally, we evaluated city-level heterogeneity using the Cochran Q test and $I^2$ statistic for each univariable meta-regression model and multivariable meta-regression model[62]. Detailed descriptions of the two-stage analytical

framework and specific methodologies can be found in Supplementary Method.

## Estimation of attributable fractions

Utilizing a method that allows for more precise estimations of disease burden by accounting for the potentially intricate lag patterns of CEHWI-related mortality risks[63], we quantified the mortality burden (attributable deaths) due to CEHWI for each city based on their best linear unbiased predictions (BLUPs) of the cumulative exposure-response curves[51,61]. The BLUPs strike a balance between the city-specific curves derived from the first-stage analysis and the pooled results, enabling cities with shorter data series to borrow strength from those with larger sample sizes and similar characteristics. The AF was computed by dividing the number of attributable deaths by the total deaths in the summer seasons. We subsequently derived 95% eCIs through Monte Carlo simulations, based on the assumption of a multivariate normal distribution for the BLUP of the estimated coefficients. The detailed procedures for these calculations are provided in Supplementary Method.

## Sensitivity analyses

To ensure the robustness of our findings, we performed a series of sensitivity analyses. First, we included daily average concentrations of $PM_{10}$ and $O_3$ into the main model, both individually and simultaneously, to assess the potential confounding effects of air pollution. The sensitivity analysis involving $O_3$ was confined to 13 cities with complete air pollution data throughout the study period. Second, we changed parameters in the main models as per previous studies[8,15,58,62]. These adjustments encompassed: *dfs* for relative humidity (ranging from three to five), the cross-basis function of CEHWI in the exposure-response dimension (redefining the function by a natural cubic B spline function as opposed to a natural spline function), and the knots for the cross-basis function (using three and four internal knots at equally spaced percentiles on the log scale, respectively). Thirdly, to evaluate whether the 90th percentile threshold adequately represents extreme heat, we fitted exposure-response relationships between summer temperatures and mortality, comparing the MMT with the 90th percentile threshold. Additionally, to account for the impact of varying heatwave thresholds and durations, we altered the threshold from the 90th percentile to the 92.5th, 95th, and 97.5th percentiles, and adjusted the duration from two days to three and four days, respectively. To verify that the observed elevated impacts of CHW were not driven by the form of the CEHWI, we transformed heatwaves into traditional binary variables and reran the models.

All statistical analyses were performed using R (version 4.0.3; R Project for Statistical Computing) with the packages "dlnm" (version 2.4.7) and 'mvmeta' (version 1.0.3). A two-tailed *P* value of <0.05 was considered statistically significant.

## Reporting summary

Further information on research design is available in the Nature Portfolio Reporting Summary linked to this article.

## Data availability

The hourly temperature observations for all selected cities in this study were extracted from the ERA5-Land database (https://cds.climate.copernicus.eu). Relative humidity data were obtained from the Korean Meteorological Office (https://www.kma.go.kr/eng/index.jsp), the Japanese Meteorological Agency (https://www.jma.go.jp/jma/indexe.html), and the Chinese Meteorological Data Sharing Service System(https://data.cma.cn/en). Daily concentrations of $PM_{10}$ and $O_3$ were downloaded from the Korean Research Institute of Public Health, the Japanese National Institute for Environmental Studies (https://www.nies.go.jp/index-e.html), and the Environmental Monitoring

Centre of China (https://www.cnemc.cn/en/). City-specific attributes were obtained from Statistics Korea (https://kostat.go.kr/ansk/), the Statistics Bureau of Japan (https://www.stat.go.jp/), the National Bureau of Statistics of China (https://www.stats.gov.cn/sj/ndsj/), and the Census and Statistics Department of Hong Kong (https://www.censtatd.gov.hk/). Due to restrictions imposed by the data owners, the daily mortality data for each city of this study cannot be publicly released. They are only available upon request from the corresponding author and other relevant authors. Source data are provided with this paper.

## Code availability

The code repository for the main statistical analysis is available at: https://github.com/Simon-JD-Liu/East_Asia_heatwave[64].

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

## Acknowledgements

This work was supported by the National Natural Science Foundation of China (82430105) (H.K.), Shanghai B&R Joint Laboratory Project (22230750300) (R.C. and H.K.), Shanghai Municipal Science and Technology Major Project (2023SHZDZX02) (R.C. and H.K.), Shanghai International Science and Technology Partnership Project (21230780200) (R.C. and H.K.), and the Key Project for High-quality Development of Public Health Co-operated by Fudan University and Jiading District (GWGZLXK-2023-03) (R.C.).

## Author contributions

H.Kan and R.C. contributed to the study conceptualisation. J.L. coordinated the work, curated the exposure data, conducted the formal analysis, created the visualizations, drafted the original manuscript, and edited the final version. H.Kim, M.H., W.L., Y.H., S.E.K., H.Kan, and R.C. contributed to the collection of the mortality database, the interpretation of the results, and the manuscript revision. C.H. contributed to the manuscript revision. H.Kan and R.C. supervised all data analysis and manuscript preparation, as well as secured funding. The corresponding authors (H.Kan and R.C.) had full access to all the data in the study and had final responsibility for the decision to submit for publication after obtaining approval from J.L., H.Kim, M.H., W.L., Y.H., S.E.K., and C.H.

## Competing interests

The authors declare no competing interests.
