## [Peer Review File · Nature Communications]

REVIEWER COMMENTS

Reviewer #1 (Remarks to the Author):

Review of Liu et al, Nonlinear exposure-response associations of daytime, nighttime, and day-night compound heatwaves with mortality: a multi-country study in East Asia

This paper returns to a much-studied and important subject, the relationship between extreme heat and human mortality. Many studies have clearly demonstrated associations between extreme heat and mortality. However, very few studies have analyzed this relationships in the context of the temporal behavior of extreme heat. One interesting temporal feature is compounding, a result of the sequencing of extremes that has been cited as having the potential to increase mortality rates as climate warms (Wang, J, Nature Clim Change, 2021; Baldwin et al, Earth's Future, 2019). Based on an analysis of the empirical records of heat and mortality, Liu et al may be the first multinational study to demonstrate a significant difference in heat related mortality rates based on sequencing. They focus specifically on data from 28 cities in South Korea, Japan, and China during 1981-2010, comparing heat/mortality relationships for hot-day-cool-night, cool-day-hot-night, and hot-day-hot-night sequences, the latter being a compound extreme event. They find notably higher mortality rates in compound heat events than in the other two sequences, which is interesting not only for its future implications but because it is consistent with the expectation that a nighttime respite from heat is critical to recovery from daytime heat stress.

Liu et al hued closely to analysis methods that are standard in heat/mortality studies. The aspect of their approach that may be novel in the current context and is critical to the results is their structuring of the analysis as a threshold problem where the treatment variable is not temperature, but an index based on cumulative time above a locally defined threshold. Regrettably, the paper is opaque on this critical aspect, referring to an earlier paper by other authors and giving a skimpy explanation of the relationships that define the index in Methods (lines 367-381) without physical justification and no explanation or justification at all in the main text. That the main analysis is primarily based on one choice of heat wave duration and sequencing when an infinite number are theoretically possible makes it especially important that the reader be able to understand why these choice were made beyond simply referring to someone else's analysis. To some extent, this problem is addressed by the sensitivity analysis. However, the reader still needs 1) a more comprehensive demonstration that the effect of compounding isn't a fluke outcome of the particular choices of duration, sequencing of hot periods (contiguously), and form of the index, and 2) some intuition as to why the specific choices made in defining the index and thresholds make sense.

In addition, I am troubled by the lack of any significant air pollution effect, an outcome that flies in the face of other studies showing combined effects of heat and pollution on risk. Could it be that the effects of air pollution and heat are sufficiently correlated that the effect of pollution is absorbed by the effect of extreme heat? Finally, I'd like to see a discussion of the quality of the mortality data, at least by reference to other studies.

Overall, I would recommend a major revision. Given that other smaller-scale studies have made a similar point, it is imperative that this study be sufficiently clear so as to be accessible to a relatively wide audience.

Reviewer #2 (Remarks to the Author):

Overarching comments

This is an interesting piece of research showing the impacts of compound day-night heat on mortality. The authors employ a new “Cumulative Excess Heatwave Index” (CEHWI) to describe the anomaly of heatwave intensity. They find this index identifies a nonlinear relationship between heatwaves and mortality and is more useful than a binary representation.

The study also identifies the lack of human adaptability to CHW (continuously increasing excess risk, without an apparent threshold).

The authors emphasise this work is the first to consider the effects of compound day-and-night heatwaves on mortality across multiple countries (although something similar has already been done for individual countries, e.g.

<https://www.sciencedirect.com/science/article/pii/S0048969721012870>).

The lag patterns in mortality risks were also interesting to see, but look similar to published work, e.g. Figure 2 in here:

<https://www.sciencedirect.com/science/article/pii/S0048969721012870#f0010>.

Main comments

The authors describe their work as the first work treating the relationship between heatwave intensity exposure and mortality rate/risk as nonlinear (l.77, 190). From a quick search, I find other examples in the literature, e.g. <https://www.ncbi.nlm.nih.gov/pmc/articles/PMC7805588/> see the text - “The non-linear influence of temperature on the mortality rate can be quantified with the aid

of an exposure–response curve” - and Figure 1. It would be helpful to more explicitly clarify where the novelty lies, as this may be unclear to readers who are not epidemiologists.

Minor comments

L.30 Please clarify why “exposure-response curves” are described as “unexplored”.

L.42-45 It would be helpful to put these proportions in perspective with the proportions of mortality risk found elsewhere in the literature.

L.90. It would be helpful to explain and justify the choice of three types of mortality (non-accidental, cardiovascular, and respiratory) examined in the work.

L.210 “the cumulative excess heat ... takes into account the exceedance of thresholds of heatwaves, rather than merely considering the general, or absolute temperatures experienced during a heatwave”. It is unclear to me why a threshold-based approach (e.g. 90th percentile) is better than using an absolute threshold that reflects human tolerance limits to extreme heat. The 90th percentile, in many places, will not be very hot at all. I know previous studies have also frequently used this percentile-based approach but it seems illogical. Please kindly clarify this for the reader.

Related to this point, the study unveils “a discernible adaptability threshold that varies across diseases and types of heatwaves”. The thresholds also vary across regions. Could this variability be partly explained by the fact that the level of heat associated with the 90th percentile varies in different locations?

L.379 “following a prior study, we introduced the CEHWI” – it is not clear to me if the CEHWI was developed in this paper, or in that prior work. Please clarify.

L.408. Multivariate meta-regression. Could the parameters & significance testing of the models be provided?

L.438. BLUP needs defining and clarifying.

L.458. The authors conducted a number of sensitivity analyses to assess the robustness of the findings (e.g. confounders such as air pollution; choice of threshold; duration of event), however, I cannot find the results of the sensitivity testing (they are not in the appendix?).

L.480. Please provide the codes for the analysis, e.g. via a Github link. It seems similar to the approach in Zhao et al. 2024 (<https://journals.plos.org/plosmedicine/article?id=10.1371/journal.pmed.1004364>) - which they provided like this: https://github.com/qizhao89/HW_prediction/blob/main/code.r

Figures

Figures 1, 2 ,3:

- The figures are clear, but it is not straightforward to compare the lines and confidence intervals in the three columns. Perhaps representing the three lines on the same figure could help the reader see where they differ significantly? Just a thought.
- The labels “Hot-Day-Cool-Night” etc should probably be above the “(A) East Asia” label.
- It may be worth specifying how many data points are considered in each curve.
- The captions for Figures 1-2 say vertical dotted lines indicate the 90th percentile but do not mention the other percentiles.
- “Shaded areas represent”... this should be mentioned after “solid lines represent”
- Would define CEHWI in the first line of the caption, not the last.

Figure 4 is clear.

Figure 5 is clear.

Reviewer #3 (Remarks to the Author):

This study uses the Cumulative Excess Heatwave Index (CEHWI) to examine the associations between different types of heatwaves (daytime-only, nighttime-only, and compound) and mortality rates (non-accidental, cardiovascular, and respiratory) in 28 East Asian cities. The paper makes a valuable contribution by making the case for the use of continuous measures to study non-linear associations between heat and mortality. Consideration of different types of heatwaves across mortality causes is also a strength of the study. I want to offer the following suggestions for the authors' consideration.

Major Comments:

1. While the utility of the CEHWI is well-discussed in the discussion section, it would be beneficial to provide more context in the abstract and introduction. Perhaps the authors could elaborate on the CEHWI, addressing questions such as its definition, computation method, and advantages over existing measures in studying health impacts.
2. The paper touches upon short-term adaptability when discussing non-linear trends in the findings. It might be helpful to introduce the concept of adaptability earlier in the paper, along with any hypothesized mechanisms the authors considered during the study.

3. The authors note that the "adaptability threshold varies across diseases and types of heatwaves" (line 218). While the discussion provides plausible reasons for variations across heatwaves, it might be worthwhile to expand on the variations observed across diseases. For instance, it would be interesting to explore why the exposure-response curve for respiratory diseases (in China and South Korea) differs significantly from all-cause mortality or cardiovascular diseases (in the case of daytime heatwaves). The authors' insights into the potential reasons for these variations across mortality causes would be valuable.

4. Regarding the previous point, the authors state that "Consistent with existing knowledge, our results affirm the heightened vulnerability of patients with cardiovascular and respiratory diseases to heatwaves, irrespective of heatwave types" (lines 263-265). However, this statement may not hold universally, as evidenced by the case of China for daytime heatwaves (Figures 1-3). It might be worth addressing this apparent inconsistency.

5. For the lag structures of all-cause mortality for daytime heatwaves in China (Fig S1), the authors note a longer lag. It would be beneficial to discuss potential reasons for this anomaly.

6. Several anomalies are observed for China. Could this be due to the inclusion of the most northwestern city, which might have a significantly different climate? A sensitivity analysis excluding this city could provide valuable insights.

7. In explaining the observed differences between daytime and nighttime heatwaves, the authors refer to differences in daytime vs. nighttime exposures. While this is a reasonable approach, it might be prudent to acknowledge the limitations of such inferences, given the lack of information about individuals' occupations in the sample. For instance, we cannot determine whether individuals primarily worked outdoors or indoors during the day.

Minor Comments:

- Line 425: It would be helpful to specify the potential effect modifiers considered. The text mentions that details are provided in the appendix, but the specific variables are not indicated there either.

- Line 437: Consider explaining or expanding on the abbreviations ECIs and BLUP for clarity.

- Line 432: A brief description of the forward method would be beneficial for readers unfamiliar with the technique.

I hope these comments prove helpful in further strengthening this already impressive study. The authors' work on the CEHWI and its application to different types of heatwaves provides valuable insights into the complex relationship between heat exposure and mortality.

General replies to the Reviewers' comments:

We sincerely appreciate the highly constructive suggestions and comments from the editor and three referees. We have carefully revised our manuscript based on the feedback provided, which we believe has improved our manuscript a lot. We hope our revisions and responses are clear and satisfactory. If any further revisions are required, we would be happy to follow your suggestions again.

Point-by-point responses are presented below, with the referees' comments in black and our responses in blue. In the revised manuscript, all major revisions are also highlighted in blue.

Reviewer #1 (Remarks to the Author):

Review of Liu et al, Nonlinear exposure-response associations of daytime, nighttime, and day-night compound heatwaves with mortality: a multi-country study in East Asia

This paper returns to a much-studied and important subject, the relationship between extreme heat and human mortality. Many studies have clearly demonstrated associations between extreme heat and mortality. However, very few studies have analyzed this relationships in the context of the temporal behavior of extreme heat. One interesting temporal feature is compounding, a result of the sequencing of extremes that has been cited as having the potential to increase mortality rates as climate warms (Wang, J, Nature Clim Change, 2021; Baldwin et al, Earth's Future, 2019). Based on an analysis of the empirical records of heat and mortality, Liu et al may be the first multinational study to demonstrate a significant difference in heat related mortality rates based on sequencing. They focus specifically on data from 28 cities in South Korea, Japan, and China during 1981-2010, comparing heat/mortality relationships for hot-day-cool-night, cool-day-hot-night, and hot-day-hot-night sequences, the latter being a compound extreme event. They find notably higher mortality rates in compound heat events than in the other two sequences, which is interesting not only for its future implications but because it is consistent with the expectation that a nighttime respite from heat is critical to recovery from daytime heat stress.

Response: Thank you very much for your thorough and professional review. We are particularly encouraged by your recognition of our approach in comparing mortality risks for different heatwave types and highlighting the more elevated mortality risk associated with compound heatwaves, which aligns with the expectation that nighttime respite from heat is critical for recovery.

We also appreciate your reference to the publications of Wang et al. (2021) and Baldwin et al. (2019). Wang et al. (2021) had already been cited in the original version, and Baldwin et al. (2019) has now been added to the revised manuscript (*reference 27 and line 73*).

Furthermore, another key strength of our study is providing first-hand evidence on the potential nonlinear association between short-term heatwave exposure and mortality risk. This finding challenges the adequacy of using simple binary variables to define heatwaves in epidemiological studies and health risk assessments,

highlighting the need for more nuanced approaches.

Liu et al hued closely to analysis methods that are standard in heat/mortality studies. The aspect of their approach that may be novel in the current context and is critical to the results is their structuring of the analysis as a threshold problem where the treatment variable is not temperature, but an index based on cumulative time above a locally defined threshold. Regrettably, the paper is opaque on this critical aspect, referring to an earlier paper by other authors and giving a skimpy explanation of the relationships that define the index in Methods (lines 367-381) without physical justification and no explanation or justification at all in the main text. That the main analysis is primarily based on one choice of heat wave duration and sequencing when an infinite number are theoretically possible makes it especially important that the reader be able to understand why these choice were made beyond simply referring to someone else's analysis. To some extent, this problem is addressed by the sensitivity analysis. However, the reader still needs 1) a more comprehensive demonstration that the effect of compounding isn't a fluke outcome of the particular choices of duration, sequencing of hot periods (contiguously), and form of the index, and 2) some intuition as to why the specific choices made in defining the index and thresholds make sense.

Response: We greatly appreciate your highly constructive suggestions, upon which we have carefully revised our manuscript. **To ensure clarity, we have structured our response into four main sections**, which are as follows: **(1)** Justifications for the Cumulative Excess Heatwave Index (CEHWI); **(2)** Explanations for the three types of heatwaves; **(3)** Sensitivity analysis for **(a)** choices of duration, **(b)** threshold, **(c)** sequence, and **(d)** form of heatwaves; **(4)** Intuition for the choices in defining the index and threshold.

(1) Epidemiological and Physical Justifications for the Cumulative Excess Heatwave Index (CEHWI)

We totally agree with you that simply referencing prior studies does not provide sufficient justification for the use of CEHWI.

a. Epidemiological justification

Firstly, although the *Methods section* lacks a physical explanation for this index, an epidemiological rationale for its use had already been provided in the *Introduction* and *Discussion* sections of the original version (in lines 59-71 and 204-212 of the original manuscript; **now lines 64-70 and 197-201 of the revised manuscript**). Therefore, we did not repeat epidemiological explanations in the Methods section.

Briefly, prevailing heatwave-health risk assessment studies oversimplify heatwaves as binary variables (i.e., 0-1 variables), overlooking the dynamic and complicated responses of populations to heatwaves with varying intensities resulting from global warming. It is crucial to understand variations in health impacts of heatwaves across different cumulative heat scales in the context of global warming, where heatwave intensity is increasing at different rates across regions. Additionally, the well-established nonlinear relationships between temperatures and adverse health outcomes necessitate an index like CEHWI, which can capture the complex and potentially nonlinear health effects of heatwaves. Please see *lines 64-70* and *197-201* for details.

b. Physical Justification:

Inspired by your suggestions, we acknowledge the need for detailed physical justifications. The physical justifications for the index primarily include two parts:

Accumulation and Persistence of Excess Heat: Heatwave is virtually characterized by the dynamic and continuous process of extreme heat, not just merely a series of isolated high-temperature events represented by binary variables. The CEHWI captures the accumulation and persistence of excess heat over time by summing the hourly temperature differences that exceed the heatwave threshold during heatwave days. This index accounts for the ongoing thermal load (i.e., varying levels of heatwave intensities) experienced by the environment and human populations. Additionally, by focusing on the persistence of heatwaves, it provides a nuanced measure that can more accurately represent cumulative heat stress effects that build up over consecutive hours or days.

More direct physical stress and damage from heat: The CEHWI emphasizes excess heat, which defined as the temperature that exceeds a locally determined heatwave threshold. This threshold, based on 30-year local climate data, considers regional climate characteristics and the population's adaptive capacity. The focus on excess heat, rather than just the absolute temperature on heatwave days, is key because it is the degree of exceedance over the threshold, not just the overall temperatures, that is most likely to cause adverse effects^[1]. This is because populations inherently possess certain thermoregulatory and heat adaptation capabilities. By quantifying this excess, the CEHWI offers a more precise assessment of the physical stress and potential damage associated with heatwaves, which aligns with the understanding that it is the deviation from normal conditions, that drives health risks.

These physical justifications are now explicitly detailed in the Methods section.

Please see lines 399-410 or the text below:

- “We employed the CEHWI to represent the cumulative and dynamic nature of heatwave intensities, which was modified according to a cumulative heat indicator developed in a prior study³¹. This index could better capture the accumulation and persistence of heat at different levels during heatwaves, rather than simplifying heatwaves as binary variables. Additionally, it specifically focuses on the additional heat that exceeds heatwave thresholds, as it is the excess temperature beyond these thresholds that poses the greatest risk to the population’s health^{31,32}. This approach serves as a nuanced measure to explore the non-linear health impacts of heatwaves, providing a potentially more accurate assessment of the hazardous impacts of heatwaves. This index was computed as the sum of the hourly temperature difference between each heatwave within the respective periods and the calendar-day heatwave threshold, as follows:....”

2. Justifications for the Classification of Heatwaves

a. Biological Justification:

The possible biological rationale underlying the health effects of different types of heatwaves had already been mentioned in the *Discussion* section in the original manuscript (original lines 231-245; **please refer to lines 238-241 of the revised manuscript**), and not repeated in the *Methods* section. Briefly, daytime heatwaves increase the risk of cardiopulmonary events due to direct heat stress,^[2-3] while nighttime heatwaves disrupt thermoregulation and sleep,^[4-5] leading to increased mortality risks. Compound heatwaves might have a “double-strike” effect,^[6-7] because the inability to cool down at night exacerbates the stress from daytime heat, significantly increasing the risk of heat-related mortality.

b. Physical Justification:

Following your suggestion, **we have now included relevant physical justification for heatwave classification**. Briefly, the distinct physical processes that drive daytime and nighttime heatwaves—such as anticyclonic conditions^[8-10] for daytime heatwaves and insufficient nocturnal radiative cooling^[11-15] for nighttime heatwaves—justify their separate classification. Compound heatwaves typically result from the strong day-night land-air interaction process, dry soil conditions, and increased anti-cyclonic conditions,^[11, 14-16] leading to the intensification of coupling of daytime and nighttime heat.

Please see lines 379-385 in the Methods section or the text below:

- “Since the physical conditions that trigger heatwaves during different times of the day—daytime and nighttime—are distinct^{27,29,50}, the physiological responses to heat stress are also distinct. For example, daytime heatwaves are primarily caused by solar radiation and anticyclonic conditions leading to surface heating^{28,51}. In contrast, nighttime heatwaves result from insufficient nocturnal radiative cooling of accumulated daytime heat⁵², with dry soil conditions and strong day-night land-air interactions further intensifying compound heatwaves⁵³. Given these differences, we categorized heatwaves into three distinct types: (a) daytime heatwaves.....”.

3. Sensitivity Analysis for Choices of Duration, Threshold, Form, and Sequence

Following the reviewer’s suggestion, we conducted several additional analyses to ensure that the observed larger effects of compound heatwaves are not merely artifacts of the specific choices regarding *duration*, *threshold*, *form of the index*, and *sequencing of hot periods*.

a. Duration

As noted by the reviewer, in the original version, we had already performed a sensitivity analysis by adjusting the duration **from at least *two days to three and four days (lines 488-491)***, respectively. According to the WHO, a heatwave is defined as an extreme temperature event lasting for at least two or three days, making these durations the most common settings in the literature. Some studies also explore longer heatwave effects by setting the duration to at least four days. Therefore, the sensitivity analysis ranging from two to four days can validate the impact of duration settings. Since the commonly accepted definitions of heatwaves in the literature require them to last at least two days, we did not change the duration to one day, as doing so would violate this established definition.

b. Threshold

Similarly, our previous sensitivity analysis **had already** altered the threshold **from the *90th percentile to the 92.5th and 95th percentiles*** of historical temperature distributions. We have **now added** an analysis using the ***97.5th percentile*** to further explore the impact of different thresholds. We did not consider the 99th percentile due to the almost complete absence of identified heatwaves under such a strict threshold.

These analyses consistently show that compound heatwaves have a significantly higher effect on mortality than daytime or nighttime heatwaves, **regardless of the varied definitions**. (See *Supplementary Table 5* or *the Table at the end of this*

response for details.)

We have now added the following sentences:

- **Methods** (Lines 488-490): “To account for the impact of varying heatwave thresholds and durations, we altered the threshold from the 90th percentile to the 92.5th, 95th, **and 97.5th** percentiles, and the duration from two days to three and four days, respectively.”
- **Results** (Lines 176-178): “With increasing thresholds and duration in the heatwave definition, the mortality risks of heatwaves showed an upward trend. **Nonetheless, the risks related to CHW were consistently higher than those to DHW and NHW (Supplementary Table 5).**”

c. Form of the index

We also examined the impact of the index's form by degrading heatwaves from CEHWI to a traditional binary heatwave form and reran the models. The results indicated a slight reduction in effect sizes for all types of heatwaves, but the higher impact of compound heatwaves persisted, confirming that the observed higher effects of compound heatwaves are not merely artifacts of the index's form.

We have now added:

- **Methods** (Lines 491-493): “**To verify that the observed higher effects of CHW were not driven by the form of the CEHWI, we transformed heatwaves into traditional binary variables and reran the models**”
- **Results** (Lines 179-180): “**For the binary form of heatwaves, the effect of CHW remained higher than that of DHW and NHW (Supplementary Table 5)**”

d. Sequencing of hot periods

We realized that our previous description using terms like “hot-day-hot-night” may have led to some misunderstandings. To clarify, the definitions of heatwaves in our study do **not** inherently distinguish between the order of daytime and nighttime heatwaves, but instead focus on the temporal structure of heatwaves. Specifically, compound heatwaves (CHW) are characterized by at least two consecutive days where both daytime and nighttime temperatures exceed their respective 90th percentiles. This definition naturally includes the sequences that alternate between hot nights and hot days (i.e., hot night–hot day–hot night–hot day...; see the following

paragraph for why this sequence is used). Thus, both hot-night-hot-day and hot-day-hot-night sequences are included under our CHW category.

Specifically, given that each day's timing begins at 00:00 (midnight), the definition logically follows a sequence from night to day. Technically, a “hot-day-hot-night” sequence spans across two days, beginning with a day (sunrise of the day) and continuing into the following night (ending at sunrise the next day). However, this sequence cannot be directly analyzed in relation to our outcome variable (daily death records), **as epidemiological studies require that exposure (in this case, heatwaves) occurs before the outcome**, which is collected up to 24:00 of each day. Therefore, for any given day, the only feasible sequence is “night to day”, rather than a cross-day “day to night” sequence. Future studies with hourly mortality and morbidity data can further thoroughly explore the risk differences between these two types of compound heatwaves, while the aim of this study is to investigate the overall effects of compound heatwaves without further subdivision.

Nevertheless, it is unlikely that the sequence of hot periods is the primary driver of the observed higher effects for compound heatwaves. As mentioned earlier, by requiring heatwaves to last for at least two consecutive days, our analysis inherently captures any potential sequence (hot nights followed by hot days, and hot days followed by hot nights). Additionally, our distributed lag model accounts for heatwave exposure up to one week before the mortality event (i.e., a lag of one week; see lines 427-428 for details), ensuring that the cumulative effects of heatwaves—regardless of their internal sequence—are fully captured in the analysis. As a result, **the analysis does not prioritize any specific sequence over another**, mitigating concerns that the results might be skewed by the order of hot periods within the compound heatwave.

Lastly, we would like to emphasize that the internal sequence within each type of heatwave (daytime, nighttime, and compound) does not affect the effect estimates of other two types, as the reference category remains normal days. Specifically, these heatwaves are treated as a four-category variable, modeled using three dummy variables, with “normal days” (without any heatwaves) as the reference category.

To avoid further confusion, **we have now avoided using terms like “hot-day-cool-night”, “cool-day-hot-night”, and “hot-day-hot-night”, but instead employed more precise and commonly used terminology: “daytime-only heatwaves (briefly as daytime heatwaves)”, “nighttime-only heatwaves (briefly as nighttime heatwaves)”, and “compound heatwaves” throughout the revised manuscript.** (The term change was not highlighted in blue in the revised manuscript to avoid excessive marking, which might hinder the review of other modifications).

Overall, with the above revisions and sensitivity analyses, the higher mortality risks associated with compound heatwaves appear **not to be a consequence of specific sequencing, duration and threshold choices, or the newly employed form in this study**. Instead, the higher risks are likely driven by the cumulative stress of consecutive hot days and nights, which fail to provide the human body with sufficient relief or recovery time.

4. Intuition Behind the Choices for the Index and Thresholds

Regarding the index, our choices centered on calculating cumulative excess heat rather than just the specific temperatures during heatwave days, as well as determining the appropriate duration and threshold.

For calculating cumulative excess heat (i.e., form of the index), as previously mentioned, it provides a nuanced measure for capturing the potential non-linear health impacts of heatwaves and offers a more precise assessment of their hazardous effects (please refer to *lines 399-407* for details).

Regarding the choice of a minimum two-day duration, this is in line with the guidelines on “*Heatwaves and Health*” jointly published by the WHO and the World Meteorological Organization (WMO).^[17] Additionally, it is one of the most widely used duration settings in current literatures.

As for the choice of the 90th percentile threshold, there are two main reasons (*lines 391-393 and 373*):

- **Balancing extremity and sample size:** The 90th percentile strikes an appropriate balance between event extremity and the sample size needed for meaningful analysis.^[6-7] It effectively captures the potential for relatively moderate extremes to result in significant health impacts.
- **Consistency with established practices:** The use of the 90th percentile is consistent with the common practice in both climate science and epidemiology, aligning our study with prior researches in this field.^[6-7, 18-25]

Supplementary Table 5. The pooled relative risks of daily non-accidental, cardiovascular, and respiratory mortality associated with daytime, nighttime, and day-night compound heatwave across East Asia, under different thresholds, durations,

and the form of heatwave.

Heatwave	Types	NAD	CVD	RD
Changing Threshold (from 90th to 92.5th, 95th, and 97.5th, respectively)				
HWT_{92.5}_D₂				
	DHW	1.11 (1.07, 1.15)	1.06 (1.04, 1.09)	1.07 (1.02, 1.12)
	NHW	1.03 (1.02, 1.05)	0.99 (0.95, 1.04)	1.03 (1.00, 1.05)
	CHW	1.21 (1.14, 1.29)	1.32 (1.21, 1.45)	1.12 (1.08, 1.17)
HWT₉₅_D₂				
	DHW	1.03 (1.00, 1.06)	1.12 (1.04, 1.20)	1.01 (0.92, 1.12)
	NHW	1.02 (0.97, 1.07)	1.13 (1.04, 1.22)	0.98 (0.94, 1.03)
	CHW	1.26 (1.20, 1.32)	1.44 (1.30, 1.59)	1.21 (1.16, 1.26)
HWT_{97.5}_D₂				
	DHW	1.00 (0.95, 1.04)	1.06 (0.93, 1.22)	1.04 (0.88, 1.23)
	NHW	1.04 (1.00, 1.08)	1.05 (0.96, 1.15)	1.05 (0.96, 1.15)
	CHW	1.32 (1.20, 1.45)	1.57 (1.41, 1.74)	1.18 (1.07, 1.30)
Changing Duration (from at least 2 days to 3 and 4 days, respectively)				
HWT₉₀_D₃				
	DHW	1.01 (0.96, 1.06)	1.04 (0.96, 1.12)	0.99 (0.95, 1.03)
	NHW	1.01 (1.01, 1.02)	0.97 (0.94, 0.99)	1.01 (0.98, 1.04)
	CHW	1.22 (1.16, 1.29)	1.38 (1.26, 1.52)	1.25 (1.14, 1.38)
HWT₉₀_D₄				
	DHW	1.07 (1.04, 1.11)	1.08 (0.98, 1.19)	0.86 (0.73, 1.02)
	NHW	1.03 (1.02, 1.05)	0.96 (0.93, 1.00)	0.98 (0.94, 1.02)
	CHW	1.27 (1.21, 1.34)	1.49 (1.39, 1.60)	1.23 (1.09, 1.39)
Changing Forms of heatwaves (from CEHWI to a 0-1 variable)				
HWT₉₀_D₂*				

DHW	1.03 (1.03, 1.04)	1.04 (1.02, 1.06)	1.05 (1.01, 1.08)
NHW	1.01 (1.00, 1.02)	0.99 (0.96, 1.02)	1.02 (1.00, 1.03)
CHW	1.13 (1.10, 1.16)	1.18 (1.16, 1.21)	1.10 (1.08, 1.11)

*Heatwaves were defined based on traditional binary variables.

DHW=daytime heatwave; NHW=nighttime heatwave; CHW=day-night compound heatwave;
NAD=non-accidental death; CVD=cardiovascular disease; RD=respiratory disease.

Reference:

- [1] Perkins-Kirkpatrick SE, Lewis SC. Increasing trends in regional heatwaves. *Nat Commun* 2020; 11(1): 3357.
- [2] Yang J, Zhou M, Ren Z, et al. Projecting heat-related excess mortality under climate change scenarios in China. *Nat Commun* 2021; 12(1): 1039.
- [3] Guo Y, Gasparrini A, Li S, et al. Quantifying excess deaths related to heatwaves under climate change scenarios: A multicountry time series modelling study. *Plos Med* 2018; 15(7): e1002629.
- [4] Okamoto-Mizuno K., et al. Effects of humid heat exposure on human sleep stages and body temperature. *Sleep*. 1999; 22: 767–73.
- [5] He C, Kim H, Hashizume M, et al. The effects of night-time warming on mortality burden under future climate change scenarios: a modelling study. *Lancet Planet Health* 2022; 6(8): e648-57.
- [6] Wang J, Chen Y, Liao WL, et al. Anthropogenic emissions and urbanization increase risk of compound hot extremes in cities. *Nat Clim Change* 2021; 11(12): 1084.
- [7] Liu, J., Qi, J., Yin, P. et al. Rising cause-specific mortality risk and burden of compound heatwaves amid climate change. *Nat. Clim. Chang* 2024.
- [8] Horton, DE, Johnson, NC, Singh, D, et al. Contribution of changes in atmospheric circulation patterns to extreme temperature trends. *Nature* 2015; 522: 465–469.
- [9] Wang J, Chen Y, Tett S, et al. Anthropogenically-driven increases in the risks of summertime compound hot extremes. *Nat Commun* 2020; 11(1): 528.
- [10] Lee, MH, Lee, S, Song, HJ & Ho, CH. The recent increase in the occurrence of a boreal summer teleconnection and its relationship with temperature extremes. *J. Clim.* 2017; 30: 7493-7504.
- [11] Fischer, EM, Seneviratne, SI, Lüthi, D & Schär, C. Contribution of land-atmosphere coupling to recent European summer heat waves. *Geophys. Res. Lett.* 2007; 34: L06707.

- [12] Vogel, MM, Orth, R, Cheruy, F, et al. Regional amplification of projected changes in extreme temperatures strongly controlled by soil moisture-temperature feedbacks. *Geophys. Res. Lett.* 2017; 44: 1511–1519.
- [13] Donat, MG, Pitman, AJ & Angéilil, O. Understanding and reducing future uncertainty in mid-latitude daily heat extremes via land surface feedback constraints. *Geophys. Res. Lett.* 2018; 45: 10,627–10,636.
- [14] Black, E., Blackburn, M., Harrison, G., et al. Factors contributing to the summer 2003 European heatwave. *Weather* 2004; 59: 217–223.
- [15] Miralles, DG, Teuling, AJ, van Heerwaarden, CC & de Arellano, JV. Mega-heatwave temperatures due to combined soil desiccation and atmospheric heat accumulation. *Nature Geosci.* 2014; 7: 345–349.
- [16] Nairn, JR & Fawcett, RG. Defining heatwaves: heatwave defined as a heat-impact event servicing all community and business sectors in Australia. The Centre for Australian Weather and Climate Research (2013).
- [17] WMO and WHO, 2015. Heatwaves and Health: Guidance on Warning-System Development. WMO-No. 1142. World Meteorological Organization (WMO) and World Health Organization (WHO). <https://www.who.int/publications/m/item/heatwaves-and-health--guidance-on-warning-system-development>. Accessed 26th August 2024.
- [18] Wang PY, Tang JP, Sun XG, et al. Heat Waves in China: Definitions, Leading Patterns, and Connections to Large-Scale Atmospheric Circulation and SSTs. *J Geophys Res-Atmos* 2017; 122(20): 10679-99.
- [19] Russo S, Sillmann J, Fischer EM. Top ten European heatwaves since 1950 and their occurrence in the coming decades. *Environ Res Lett* 2015; 10(12): 124003.
- [20] Su Q, Dong B. Recent Decadal Changes in Heat Waves over China: Drivers and Mechanisms. *J Climate* 2019; 32(14): 4215-34.
- [21] Chen Y, Zhou B, Zhai P, Moufouma-Okia W. Half-a-Degree Matters for Reducing and Delaying Global Land Exposure to Combined Daytime-Nighttime Hot Extremes. *Earth's Future* 2019; 7: 953-66.
- [22] Russo S, Marchese A F, Sillmann J and Immé G. When will unusual heat waves become normal in a warming Africa? *Environ. Res. Lett.* 2016; 11 (05): 4016
- [23] Luo L, Zeng F, Bai G, et al. Future injury mortality burden attributable to compound hot extremes will significantly increase in China. *Sci Total Environ* 2022; 845: 157019.
- [24] He GH, Xu YJ, Hou ZL, et al. The assessment of current mortality burden and future mortality risk attributable to compound hot extremes in China. *Sci Total Environ* 2021; 777.

[25] Li Z, Hu J, Meng R, et al. The association of compound hot extreme with mortality risk and vulnerability assessment at fine-spatial scale. *Environ Res* 2021; 198: 111213.

In addition, I am troubled by the lack of any significant air pollution effect, an outcome that flies in the face of other studies showing combined effects of heat and pollution on risk. Could it be that the effects of air pollution and heat are sufficiently correlated that the effect of pollution is absorbed by the effect of extreme heat? Finally, I'd like to see a discussion of the quality of the mortality data, at least by reference to other studies.

Response: Thank you for your valuable suggestions. We appreciate the opportunity to clarify and improve our manuscript based on your insightful comments.

1. Effects of Air Pollutants

As you meticulously pointed out, our sensitivity analysis did not observe a significant *confounding effect* of air pollution (specifically PM₁₀ and O₃) on the relationship between heatwaves and mortality (*lines 165-169*). However, this result does not imply that the independent effects of air pollution, or its potential interaction with heatwaves, are absent. **It merely suggests that the confounding effect of pollutants on the heatwave-mortality relationship is likely minimal.**

Statistically speaking, even if air pollution independently influences mortality and/or interacts with heatwaves, the degree to which heatwave effects change depends on whether air pollution acts as a confounder. This is because we adjusted for air pollution in the models without including interaction terms between pollutants and heatwaves. Therefore, **this sensitivity analysis only reflects whether air pollution confounds the heatwave effects, but not whether it acts as an effect modifier (i.e., through interaction) or its presence of independent effect on mortality.** Similar findings—that air pollution is not a major confounder of heat-related effects—have been reported in several other epidemiological studies on temperature and heat exposure.^[1-5]

2. Correlation between air pollution and heat

We also estimated the *Spearman correlation coefficients* between the three types of heatwaves and concentrations of air pollutants during the study period (**summer seasons**). The results indicated that the absolute values of correlation coefficients between the three types of heatwaves and the two pollutants **were generally below**

0.1 (please see *Table X1* below). This suggests that the correlation between air pollution and heatwaves is minimal, and **the potential absorption of air pollution effects by extreme heat in this study is unlikely.** Thus, the observed heatwave effects appear to be robust and not significantly confounded by air pollution levels.

Overall, to avoid any misunderstanding, we have revised the sentence regarding the sensitivity result of air pollution. ***Please refer to lines 165-169 or the text below:***

- “The pooled exposure-response curves between the three types of heatwaves and mortality remained largely unchanged after controlling for concentrations of particulate matter with an aerodynamic diameter $\leq 10 \mu\text{m}$ (PM₁₀) and ozone (O₃) (Supplementary Figs. 3–5), **suggesting that these pollutants did not significantly confound the relationship between heatwaves and mortality.**”

3. Quality of the Mortality Data

The present study is based on a well-established dataset in East Asia, with mortality data collected, managed, and controlled by government agencies from each country or region. These agencies include the *Korea National Statistics Office* in South Korea, the *Ministry of Health, Labor, and Welfare* in Japan, the *Census and Statistics Department* in Hong Kong, and the *local Centers for Disease Control and Prevention* in mainland China. These governmental bodies maintain strict oversight and quality control measures, ensuring the reliability and validity of the mortality data used in our analysis.

We have provided descriptions of the mortality data quality and cited relevant references in the manuscript. ***Please refer to lines 344-349 or the text below:***

- ***“The mortality data used in our analysis were collected, managed, and controlled by government agencies in each country or region, ensuring their reliability and validity^{47,48}. We obtained daily death records from the Korea National Statistics Office in South Korea, the Ministry of Health, Labor, and Welfare in Japan, the Census and Statistics Department in Hong Kong, and the local Centers for Disease Control and Prevention in mainland China.”***

Table X1. Spearman correlation coefficients between heatwaves and air pollutant concentrations.

Index	PM ₁₀	O ₃
Daytime heatwaves	0.01	0.14
Nighttime heatwaves	-0.02	-0.06
Compound heatwaves	0.09	0.04

Note: Table X1 was not included in the supplementary material to avoid excessive length.

References:

[1] Liu, J., Qi, J., Yin, P. et al. Rising cause-specific mortality risk and burden of compound heatwaves amid climate change. *Nat. Clim. Chang* 2024.

[2] Gasparrini A, Guo Y, Hashizume M, et al. Mortality risk attributable to high and low ambient temperature: a multicountry observational study. *Lancet* 2015; 386(9991): 369-75.

[3] Sun S, Weinberger KR, Nori-Sarma A, et al. Ambient heat and risks of emergency department visits among adults in the United States: time stratified case crossover study. *BMJ* 2021; 375: e65653.

[4] Chen RJ, Yin P, Wang LJ, et al. Association between ambient temperature and mortality risk and burden: time series study in 272 main Chinese cities. *BMJ* 2018; 363:k4306.

[5] Yang J, Zhou M, Ren Z, et al. Projecting heat-related excess mortality under climate change scenarios in China. *Nat Commun* 2021; 12(1): 1039.

[6] He C, Kim H, Hashizume M, et al. The effects of night-time warming on mortality burden under future climate change scenarios: a modelling study. *Lancet Planet Health* 2022; 6(8): e648-57.

Overall, I would recommend a major revision. Given that other smaller-scale studies have made a similar point, it is imperative that this study be sufficiently clear so as to be accessible to a relatively wide audience.

Response: Thank you very much for your above constructive and insightful suggestions. We have carefully revised our manuscript according to your comments, including taking special care to improve the clarity of our arguments and the logical flow of the manuscript to ensure it can be understood by a wide range of readers, including those who may not be epidemiologists. In addition, the sensitivity analyses we conducted further validate the robustness of our conclusions.

Regarding the novelty, we would like to further highlight that, aside from being the first multi-country study to explore the effects of compound heatwaves, **the main**

novelty of our research lies in examining the potential non-linear effects of heatwaves. Our study is the **first** to establish the exposure-response curves for heatwave-mortality relationships, providing the first-hand evidence of the potential nonlinearity in the association between short-term heatwave exposure and mortality risk. These findings underscore the importance of considering the complex health responses and adaptive patterns of populations to different types of heatwaves, as well as their varying degrees of cumulative intensity in future epidemiological studies, disease burden assessments, and the development of heatwave-health warning systems amid global warming. *Please refer to lines 54-70, 183-186, 196-220, and 292-297 for details.*

Therefore, we believe this revised version is now clearer and more accessible to a broader audience, and hope our revisions are satisfactory.

Reviewer #2 (Remarks to the Author):

Overarching comments

This is an interesting piece of research showing the impacts of compound day-night heat on mortality. The authors employ a new “Cumulative Excess Heatwave Index” (CEHWI) to describe the anomaly of heatwave intensity. They find this index identifies a nonlinear relationship between heatwaves and mortality and is more useful than a binary representation.

The study also identifies the lack of human adaptability to CHW (continuously increasing excess risk, without an apparent threshold).

The authors emphasise this work is the first to consider the effects of compound day-and-night heatwaves on mortality across multiple countries (although something similar has already been done for individual countries, e.g. <https://www.sciencedirect.com/science/article/pii/S0048969721012870>).

The lag patterns in mortality risks were also interesting to see, but look similar to published work, e.g. Figure 2 in here: <https://www.sciencedirect.com/science/article/pii/S0048969721012870#f0010>.

Response: Thank you very much for your thorough review and insightful suggestions. We also greatly appreciate your positive recognition of the key novelties in our study, including the identification of the nonlinear relationship between heatwaves and mortality and being the first multinational study estimating health effects of compound heatwaves. We agree with your point that smaller-scale studies have explored the impact of compound heat, but we believe that **our study's novelty remains robust for the following three main reasons:**

1. Another Key Novelty–Nonlinear Heatwave-Mortality Association:

As you kindly pointed out, the major strength of our study is the identification of a potential nonlinear association between heatwave exposure and mortality risk. This is central to our work and challenges the current use of binary variables to define heatwaves in epidemiological research and health risk assessments. Our results underscore the need to consider the complex health responses and adaptive patterns of populations to the varying degrees of cumulative intensity of heatwaves, especially under global warming. These findings have important implications for future heatwave-risk prediction research, disease burden assessments, as well as the development of heatwave-health warning systems. No previous epidemiological

research, to our knowledge, has specifically addressed this. **Please refer to *lines 54-70* in the Introduction, *lines 100-117* in the Methods, and *lines 183-186, 196-220, and 292-297* in the Discussion for details.**

2. Different Exposure Assessment:

While the referenced study (*He, et al.*) analyzed the health impact of compound *hot days* (rather than *heatwaves*), our research extends this by considering both the intensity and duration of *heatwaves*. This is crucial as *heatwaves* generally accumulate greater health risks due to their prolonged and extreme nature compared to high-temperature days alone,^[1-3] particularly concerning cardiopulmonary diseases.

Additionally, most studies on cardiopulmonary impacts have focused on *compound high temperatures* rather than *compound heatwaves*, which present distinct epidemiological risks despite their meteorological similarities. To avoid confusion for readers who may not be epidemiologists, we did not emphasize this difference in the manuscript.

3. Higher Representativeness:

In epidemiological research, representativeness is a critical factor in assessing the significance of a study, as it determines the generalizability of the findings and their applicability to regional policy-making. Our study is the first multicountry study on the health impacts of compound heat/heatwaves, providing broader representativeness and stronger evidence compared to single-country studies. Although representativeness is not inherently a novelty, it is one of the most important criterion for evaluating the significance of all epidemiological studies.

Regarding the similarity in lag patterns between our study (*Figure 4*) and those reported by *He, et al.*, this actually reflects the fact that the lagged effects of heat are generally consistent across studies. Previous research on high temperatures and *heatwaves* has also reported similar lag patterns,^[4-7] with the effects typically peaking on the day of exposure and lasting for about three to four days before diminishing. Thus, despite differences in exposure types between our study and *He, et al.*, both studies deal with heat, so it's reasonable to expect no significant differences in lag patterns. The reason we included the lag structure for the three types of *heatwaves* is primarily to demonstrate that our selection of a one-week lag is sufficient to capture the lagged effects of *heatwaves* and to compare whether the lag effects differ among the three types.

Reference:

- [1] Liu, J., Qi, J., Yin, P. et al. Rising cause-specific mortality risk and burden of compound heatwaves amid climate change. *Nat. Clim. Chang* 2024.
- [2] Cheng J, Xu Z, Bambrick H, et al. Cardiorespiratory effects of heatwaves: A systematic review and meta-analysis of global epidemiological evidence. *Environ Res* 2019; 177: 108610.
- [3] Yin P, Chen R, Wang L, et al. The added effects of heatwaves on cause-specific mortality: A nationwide analysis in 272 Chinese cities. *Environ Int* 2018; 121(Pt 1): 898-905.
- [4] Sun S, Weinberger KR, Nori-Sarma A, et al. Ambient heat and risks of emergency department visits among adults in the United States: time stratified case crossover study. *BMJ* 2021; 375: e65653. (See Figure 3)
- [5] Arisco NJ, O Sewe M, Bärnighausen T, Sié A, Zatre P, Bunker A. The effect of extreme temperature and precipitation on cause-specific deaths in rural Burkina Faso: a longitudinal study. *Lancet Planet Health* 2023; 7: e478–89. (See supplementary Figure 6A and 6B)
- [6] Chen RJ, Yin P, Wang LJ, et al. Association between ambient temperature and mortality risk and burden: time series study in 272 main Chinese cities. *BMJ* 2018; 363:k4306. (See Figure 3)
- [7] Fu SH, Gasparrini A, Rodriguez PS, Jha P. Mortality attributable to hot and cold ambient temperatures in India: a nationally representative case-crossover study. *PLoS Med* 2018; 15(7):e1002619. (See Figure 3)

Main comments

The authors describe their work as the first work treating the relationship between heatwave intensity exposure and mortality rate/risk as nonlinear (1.77, 190). From a quick search, I find other examples in the literature, e.g. <https://www.ncbi.nlm.nih.gov/pmc/articles/PMC7805588/> see the text - “The non-linear influence of temperature on the mortality rate can be quantified with the aid of an exposure–response curve” - and Figure 1. It would be helpful to more explicitly clarify where the novelty lies, as this may be unclear to readers who are not epidemiologists.

Response: Thank you for your thoughtful comments. We appreciate the opportunity to clarify the novelty of our work in this part.

1. Difference Between Heatwaves and Temperature in Epidemiological Studies

Firstly, it is important to distinguish between high temperature days and

heatwave days in epidemiological studies, as **they are treated differently in assessing health risks**. High temperature days refer to days of daily mean temperature exceeding a threshold, whereas heatwave days are extreme heat events defined based on a combination of specific thresholds and duration. High temperature is treated as *a continuous variable*,^[1-5] with its health impacts reported by exposure-response curves. In contrast, heatwave is often simplified as a *binary variable* (presence or absence of a heatwave),^[6-10] with its relationship with mortality been reported using relative risks or odds ratios (i.e., **just a point estimate with 95% CI**). The binary nature of heatwave variables miss ample information about heatwave intensity and duration (from a data perspective), limiting the ability to explore their more complex and potentially non-linear exposure-response relationships.

2. Non-linear Heatwave-Mortality Relationship

Therefore, considering the well-established temperature-mortality curves, our study hypothesizes that heatwaves could also have a non-linear relationship with mortality. We propose that the health impacts of heatwaves vary with cumulative heat stress and exposure patterns during heatwave events. Based on this hypothesis, we calculated the Cumulative Excess Heatwave Index (CEHWI) to capture the continuous and fluctuating nature of cumulative heat during heatwaves, allowing us to flexibly explore the heatwave-mortality relationship as a non-linear exposure-response curve, **which had not been done before**. This curve offers a more detailed understanding of how varying levels of cumulative heat stress during heatwaves affect mortality risk.

3. Addressing the Reference

Regarding the literature you referenced, we respectfully point out that it focuses on the non-linear relationship between **high temperatures (not heatwaves)** and mortality. **The referenced literature directly states:** “*The exposure–response curve estimated in the statistical model showed a clear association of both the mean temperature of the week in question and the mean temperature of the week before with the weekly mortality rate (figure 1).*”. Therefore, the exposure-response curve they discussed is a temperature-mortality curve, which differs from the heatwave-mortality curve we investigated in our study. **To our knowledge, our study is the first to explore the exposure-response curves of relationship specifically between heatwaves and mortality.**

4. Revised Clarification in the Manuscript

Based on your suggestion, we recognize that our previous description may have been unclear to readers who are not epidemiologists. To address this, we have revised the *Introduction* to more explicitly describe the novelty of our work in the context of the well-established temperature-mortality relationships. ***The revised text in the Introduction now reads as follows:***

- **Lines 50-60:** evidence suggests that populations could exhibit adaptive physiological responses to elevated temperatures¹⁵⁻¹⁹, such as vasodilation, increased plasma volume, and higher sweat rates. These short-term, though bounded, physiological adaptations may lead to nonlinear responses to heatwaves with varying cumulative heat intensities or distinct heat exposure patterns. Given these adaptive capacities and the knowledge that the health effects of high temperature are not strictly linear in many prior studies²⁰⁻²³, we hypothesize that the relationship between heat anomalies during heatwaves and health outcomes is not necessarily linear. However, unlike non-optimal temperature as a continuous variable²⁰⁻²³, prevailing risk assessment studies consistently oversimplified heatwaves as a binary variable⁸⁻¹⁴, which fails to capture these complex associations of heatwaves with mortality.
- **Lines 64-70:** Recognizing the limitations of the prevailing binary definition, our study calculated the Cumulative Excess Heatwave Index (CEHWI), which measures cumulative excess heat during heatwaves relative to a predefined threshold (detailed calculation described in Methods). This index captures both the fluctuating magnitudes and continuous nature of cumulative excess heat during heatwaves, thus providing a more sophisticated lens on the complex and possible nonlinear relationships in the health effects of heatwaves.

Reference:

- [1] Gasparrini A, Guo Y, Hashizume M, et al. Mortality risk attributable to high and low ambient temperature: a multicountry observational study. *Lancet* 2015; 386(9991):369-75.
- [2] Burkart KG, Brauer M, Aravkin AY, et al. Estimating the cause-specific relative risks of non-optimal temperature on daily mortality: a two-part modelling approach applied to the Global Burden of Disease Study. *Lancet* 2021; 398(10301):685-697.
- [3] Chen RJ, Yin P, Wang LJ, et al. Association between ambient temperature and mortality risk and burden: time series study in 272 main Chinese cities. *BMJ* 2018; 363:k4306.
- [4] Sun S, Weinberger KR, Nori-Sarma A, et al. Ambient heat and risks of emergency department visits among adults in the United States: time stratified case crossover study. *BMJ* 2021; 375:e65653.

[5] Ballester J, Quijal-Zamorano M, Méndez Turrubiates RF, et al. Heat-related mortality in Europe during the summer of 2022. *Nat Med* 2023; 29(7):1857-1866.

[6] Nori-Sarma A, Milando C, Weinberger KR, et al. Association Between the 2021 Heat Wave in Portland, Oregon, and Seattle, Washington, and Emergency Department Visits. *JAMA* 2022; 328(23):2360-2362.

[7] Xi D, Liu L, Zhang M. et al. Risk factors associated with heatwave mortality in Chinese adults over 65 years. *Nat Med* 2024; 30: 1489–1498.

[8] Liu J, Varghese BM, Hansen A, et al. Heat exposure and cardiovascular health outcomes: a systematic review and meta-analysis. *Lancet Planet Health*. 2022; 6(6):e484-e495.

[9] Alho AM, Oliveira AP, Viegas S, Nogueira P. Effect of heatwaves on daily hospital admissions in Portugal, 2000-18: an observational study. *Lancet Planet Health*. 2024 May;8(5):e318-e326.

[10] Liu, J., Qi, J., Yin, P. et al. Rising cause-specific mortality risk and burden of compound heatwaves amid climate change. *Nat. Clim. Chang* 2024.

Minor comments

L.30 Please clarify why “exposure-response curves” are described as “unexplored”.

Response: Thank you for your valuable comment. As we previously mentioned, there is a distinction between risk assessment for high temperature and heatwaves in epidemiological studies. Although the exposure-response curves between non-optimum temperature or high temperature and mortality have been well-established, the exposure-response curve for relationships between heatwaves and mortality has not been explored due to the binary nature of traditional heatwave definitions.

Following your suggestion, we have revised the sentence to be more precise (please refer to **lines 29-30** or the text below):

*“Heatwaves are commonly simplified as a binary variable in epidemiological studies, leaving heatwave-mortality exposure-response curves **insufficiently established.**”*

L.42-45 It would be helpful to put these proportions in perspective with the proportions of mortality risk found elsewhere in the literature.

Response: Thank you for this insightful suggestion. We understand the importance of comparing our findings with those reported in other literature. However, due to the

strict word limit for the abstract, we are unable to include a detailed comparison in this section. Instead, a more appropriate and common approach is to provide these comparisons in *the Discussion section*.

Therefore, **we have added the following content to the Discussion:**

Lines 204-212: “Our estimated disease burden is relatively lower compared to those from studies using a binary approach^{13,30}. For example, two studies in Chinese cities reported attributable fractions (AFs) of 0.96% and 1.23% for compound heat-related non-accidental mortality, respectively, higher than our estimate (0.31%). This discrepancy likely arises from our use of exposure-response curves to estimate the burden, which considers population adaptation to lower to moderate cumulative heat intensities. In contrast, binary definitions oversimplify this complex response, potentially leading to an overestimation of the burden from lower-intensity heatwaves.”

L.90. It would be helpful to explain and justify the choice of three types of mortality (non-accidental, cardiovascular, and respiratory) examined in the work.

Response: Done as suggested. Briefly, the selection of the three outcomes in our study is due to their public health relevance.

- *Non-accidental mortality:* This is one of the most commonly used endpoints in climate-related epidemiological researches. It encompasses all deaths except those due to accidents, making it a broad and inclusive measure of human deaths. Non-accidental mortality could provide a comprehensive assessment of the overall health impacts of heatwaves.
- *Cardiovascular disease (CVD):* CVD is the leading cause of death globally, accounting for approximately 28.6% of all deaths in 2021.^[1] The cardiovascular system is highly sensitive to temperature extremes, making it a critical endpoint in heatwave studies. Heat exposure can exacerbate existing cardiovascular conditions and trigger acute events such as heart attacks and strokes, making this a key area of focus in understanding the health impacts of heatwaves.
- *Respiratory disease:* Respiratory disease is also a major contributor to global mortality, responsible for around 23.1% of deaths in 2021.^[1] Temperature extremes can worsen respiratory conditions, particularly in vulnerable populations such as the elderly and those with pre-existing lung diseases. The association between traditional heatwave exposure and respiratory morbidity and mortality is well-documented, making this an essential outcome to examine.

In the *method section*, we have added the following explanation for the selection of these three outcomes. Please see lines 351-355 or the text below:

“These three mortality outcomes were selected due to their sensitivity to heat exposures and their high public health relevance^{2,49}, which ensures a comprehensive evaluation of the mortality risks associated with heatwaves by capturing both overall and specific impacts on major diseases.”

Reference

[1] GBD 2021 Collaborators. Global burden of diseases on 2021. Available at: <https://vizhub.healthdata.org/gbd-results/>. Accessed August 30, 2024.

L.210 “the cumulative excess heat ... takes into account the exceedance of thresholds of heatwaves, rather than merely considering the general, or absolute temperatures experienced during a heatwave”. It is unclear to me why a threshold-based approach (e.g. 90th percentile) is better than using an absolute threshold that reflects human tolerance limits to extreme heat. The 90th percentile, in many places, will not be very hot at all. I know previous studies have also frequently used this percentile-based approach but it seems illogical. Please kindly clarify this for the reader.

Related to this point, the study unveils “a discernible adaptability threshold that varies across diseases and types of heatwaves”. The thresholds also vary across regions. Could this variability be partly explained by the fact that the level of heat associated with the 90th percentile varies in different locations?

Response: Thank you for your valuable comments. Below, we have provided detailed responses to each of your points:

1. Rationale for Using a Relative Threshold:

➤ **Relevance of Relative Thresholds:** By using a relative threshold, such as the 90th percentile, we ensure that each city's heatwave threshold is tailored to its own historical temperature distribution, providing a more accurate reflection of local heat conditions (see the second point for why 90th percentile can represent hot days). In essence, the relative threshold serves as an absolute threshold that is specific to each city's climate condition, offering a more objective basis for comparison.

- ***Standard Practice in Multicity Epidemiological Studies:*** Using a uniform relative threshold is a standard practice in almost all multicenter epidemiological studies. Setting an optimal absolute heat threshold for each city individually would be overly subjective and impractical, since there is currently no gold standard for defining the optimal heat threshold for each city. Additionally, using a uniform absolute threshold for each city is also highly uncommon, as East Asia has significant climatological disparities across cities.

2. Validation of the 90th Percentile Threshold as a Heat Threshold

- ***Addressing Concerns About the 90th Percentile:*** We understand your concern that the 90th percentile might not represent extreme heat in all locations. However, it is important to note that **the 90th percentile threshold in our study is based on the summer temperature distribution (June to August; the hottest three months) of each year over a 30-year period, not the entire 30-year temperature distribution.** This approach ensures that the threshold reflects the right tail of the local temperature distribution, making it more representative of local heat extremes. Furthermore, the 90th percentile strikes an appropriate balance between event extremity and the sample size needed for analysis. It effectively captures the potential for relatively “moderate” extremes to result in significant health impacts.
- ***Validation of the 90th Percentile Threshold:*** To address whether the 90th percentile is indeed sufficiently hazardous across the diverse cities included in our study, we have now conducted an epidemiological analysis exploring temperature-mortality curves during summer at both regional and national levels. The results revealed that **the centiles of minimum mortality temperatures (MMT) ranged from the 0.3rd to the 41st percentile of the summer temperature distribution (see *Supplementary Fig. 6 or the Figure below*),** suggesting that the 90th percentile corresponds to a hazardous threshold in our study area. **We have added following sentences as follows:**

Methods (Lines 486-488): *“To examine the adequacy of the 90th percentile threshold in representing extreme heat, we fitted exposure-response curves for summer temperatures and mortality and compared the minimum mortality temperature (MMT) to the 90th percentile threshold.”*

Results (Lines 173-175): *“The 90th percentile of the summer temperature distribution significantly exceeds the minimum mortality temperature (MMT) (Supplementary Fig. 6), affirming its suitability as a threshold for defining extreme heat events.”*

- ***Sensitivity Analysis of Different Percentiles:*** We conducted a sensitivity analysis using stricter thresholds (92.5th, 95th, and 97.5th percentiles) to assess the robustness of our findings. The results indicated slight increases in heatwave risk estimates with higher thresholds, **but the overall conclusions** regarding the greater mortality risk and burden of compound heatwaves, compared to daytime and nighttime heatwaves, **remained consistent. Please refer to the text below for details.**

Methods (Lines 488-490): *“To account for the impact of varying heatwave thresholds..., we altered the threshold from the 90th percentile to the 92.5th, 95th, and 97.5th percentiles.....”*

Results (Lines 176-178): *“With increasing thresholds and duration in the heatwave definition, the mortality risks of heatwaves showed an upward trend. Nonetheless, the risks related to CHW were consistently higher than those to DHW and NHW (Supplementary Table 5).”*

3. Variability of Adaptability Thresholds Across Regions:

We agree with your insightful observation that the adaptability thresholds also vary across regions, which might be partly due to the differing heat levels associated with the 90th percentile used in heatwave definitions for each location. However, it is important to note that these adaptability thresholds refer to the percentiles in our exposure index where significant heat-related risks begin to emerge, and this adaptability is complex, influenced by multiple factors. In contrast, the 90th percentile in the heatwave definition is based on the long-term summer temperature distribution of each region, only aiming to account for local climate adaptation. Therefore, **the threshold used in the definition of heatwaves does not equate to the adaptability thresholds.**

In fact, this adaptability variability could more likely be influenced by distinct regional climate patterns and socioeconomic factors, even though we had controlled for these variables in our analysis (See **lines 452-454** for details: *Koppen-Geiger climate classification, gross domestic product (GDP) per capita, summer average temperature, summer temperature range, population size, latitude, and longitude*). Based on your comment, **we have expanded the discussion on variation of adaptation threshold across region:**

Lines 213-218: *“our study unveiled an additional layer of complexity in this adaptability, manifesting as a discernible adaptability threshold that varies across*

heatwave types, diseases, and regions. This variability is likely attributed to differences in health effects of distinct heatwave exposure patterns, the mechanisms by which different diseases respond to heatwaves, and variations in regional geography, climate, and socioeconomic conditions.”

Lines 276-279: “Cities with larger populations experience increased heatwave-related mortality risks, likely due to higher exposure in densely populated urban areas where the urban heat island effect is more pronounced^{43,45}.”

Supplementary Fig. 6. Cumulative exposure-response curves for associations between summer daily average temperature (°C) and non-accidental mortality in East Asia. Solid lines refer to mean estimates and shaded areas denote their 95% confidence intervals. The vertical dashed line refers to the minimum mortality

temperatures (MMT) and the temperature corresponding to the 90% percentile of local temperature distribution, respectively.

L.379 “following a prior study, we introduced the CEHWI” – it is not clear to me if the CEHWI was developed in this paper, or in that prior work. Please clarify.

Response: Thank you for pointing out the need for clarification regarding the development of the Cumulative Excess Heatwave Index (CEHWI).

The CEHWI, as used in our study, is adapted from a previous published study that focused on analyzing global heatwave trends. While we did not develop the original concept of this index, our study represents a significant advancement by applying and refining this indicator to investigate the potential non-linear health effects of heatwaves in global warming, an area that has not been sufficiently explored previously.

To avoid any potential confusion, we have revised the manuscript to better reflect the origin and adaptation of the CEHWI. We now clarify that the cumulative excess index was originally proposed in a previous study, and we have adapted it as CEHWI for use in our epidemiological analysis to explore heatwave-related health impacts. **Please refer to lines 399-401 or the text below for details:**

“We employed the CEHWI to represent the cumulative and dynamic nature of heatwave intensities, which was modified according to a cumulative heat indicator developed in a prior study³¹.”

L.408. Multivariate meta-regression. Could the parameters & significance testing of the models be provided?

Response: Done as suggested. Please refer to lines 439-442 or the text below:

“In the subsequent analysis, we employed a multivariate meta-regression approach to pool the city-specific cumulative associations between CEHWI and mortality over a week^{49,59}, which was conducted at both national (South Korea, Japan, and China) and regional (East Asia) levels. The dependent variables in the models were the three sets of reduced spline coefficients, representing the cumulative association curves across lags for each city, and the coefficient estimates were obtained using the restricted maximum likelihood estimator.”

Regarding significance testing, please see lines 450-459 or the text below:

“Considering the potential spatial heterogeneity, we employed univariable meta-regression models incorporating the following city-specific attributes separately: Koppen-Geiger climate classification, gross domestic product (GDP) per capita, summer average temperature, summer temperature range, population size, latitude, and longitude. The statistical significance of the meta-predictors in accounting for heterogeneity was assessed using the Wald test. We also fit a multivariable meta-regression model including all predictors except for latitude and longitude to avoid multicollinearity. Finally, we evaluated city-level heterogeneity using the Cochran Q test and I^2 statistic for each univariable meta-regression and the multivariable meta-regression⁶⁰.”

L.438. BLUP needs defining and clarifying.

Response: Done as requested. Please refer to *lines 464-469* or the text below:

“We quantified the mortality burden (attributable deaths) due to CEHWI for each city based on their best linear unbiased predictions (BLUPs) of cumulative exposure-response curves^{49,59}. The BLUPs strike a balance between the city-specific curves derived from the first-stage analysis and the pooled results, enabling cities with shorter data series to borrow information from cities with larger populations and similar characteristics.”

L.458. The authors conducted a number of sensitivity analyses to assess the robustness of the findings (e.g. confounders such as air pollution; choice of threshold; duration of event), however, I cannot find the results of the sensitivity testing (they are not in the appendix?).

Response: Thank you for your careful review. We have carefully double checked the main text and the appendix, and the results of all sensitivity analyses had already been included in the appendix. In the main text, **we had explicitly indicated the corresponding supplementary tables and figures where these results can be found** (please refer to *lines 164-180* for details).

For instance, regarding air pollutant, we had stated (*lines 165-169*): “*The pooled exposure-response curves between the three types of heatwaves and mortality remained largely unchanged after controlling for concentrations of particulate matter with an aerodynamic diameter $\leq 10 \mu\text{m}$ (PM_{10}) and ozone (O_3) (Supplementary Figs. 3–5), suggesting that these pollutants are not significant confounders*”.

L.480. Please provide the codes for the analysis, e.g. via a Github link. It seems similar to the approach in Zhao et al. 2024 (<https://journals.plos.org/plosmedicine/article?id=10.1371/journal.pmed.1004364>) - which they provided like this: https://github.com/qizhao89/HW_prediction/blob/main/code.r

Response: Done as required. We have revised the *Code Availability section* (**lines 519-520**) as follows: **“The code repository for the main statistical analysis is available at: https://github.com/Simon-JD-Liu/East_Asia_heatwave”.**

Please note that the code scripts will be organized into a more readable format and then uploaded at the phase of *acceptance in principle*.

Figures

Response: We greatly appreciate your constructive and thoughtful suggestions, which have significantly improved the clarity of our figures. Below we described the revisions we made based on your feedback point-by-point. To avoid our response letter too lengthy, we have only included the revised Figure 1 at the end of response. Figures 2 and 3 have been revised in a similar manner and can be seen in the revised manuscript.

Figures 1, 2 ,3:

- The figures are clear, but it is not straightforward to compare the lines and confidence intervals in the three columns. Perhaps representing the three lines on the same figure could help the reader see where they differ significantly? Just a thought.

Response: Thank you for your insightful suggestion. Following your subsequent comment, we have added data points to each curve. As a result, it is now less appropriate to combine them into a single plot. Additionally, when we attempted to merge the three columns into one, it became challenging to distinguish overlapping curves, even with different colors. **Currently, each row and column share the same axis scales (i.e., consistent X and Y axes)**, allowing for straightforward comparisons. Please see Figure 1 below or Figures 1-3 in the revised manuscript.

- The labels “Hot-Day-Cool-Night” etc should probably be above the “(A) East Asia”

label.

Response: Done as required. Please see Figure 1 below or Figures 1-3 in the revised manuscript.

- It may be worth specifying how many data points are considered in each curve.

Response: Done as required. We have now specified the number of data points (i.e., heatwave distributions) in each curve as suggested. Please refer to Figure 1 below or Figures 1-3 in the revised manuscript.

- The captions for Figures 1-2 say vertical dotted lines indicate the 90th percentile but do not mention the other percentiles.

Response: Thank you for your kind reminder. We have revised the captions to clearly indicate all vertical dotted lines. The revised caption reads: *“The vertical dotted lines from left to right indicate the 25th, 75th, and 90th percentiles of the regional cumulative excess temperature distribution of heatwaves, respectively.”* (please see *lines 680-682, 689-691, and 698-700*)

- “Shaded areas represent”... this should be mentioned after “solid lines represent”

Response: Done as required. The revised description reads: *“The solid lines represent the estimated relative risks of mortality on days with specific heatwaves compared with days without heatwaves; shaded areas represent their 95% CIs”*. (please see *lines 680, 689, and 698*).

- Would define CEHWI in the first line of the caption, not the last.

Response: Done as required. We have moved the definition of CEHWI to the first line of the caption. For example, the updated title of Fig. 1 reads as:

“Figure 1. Pooled exposure-response relationship curves of cumulative excess heatwave index (CEHWI) for daytime, nighttime, and day-night compound heatwaves with non-accidental mortality, with corresponding distributions of CEHWI”. (please see *lines 675-676, 684-685, and 693-694*).

Figure 4 is clear.

Figure 5 is clear.

Response: Thank you once again for your valuable and insightful suggestions, which have greatly enhanced the clarity and precision of our figures.

Figure 1. Pooled exposure-response relationship curves of cumulative excess heatwave index (CEHWI) for daytime, nighttime, and day-night compound heatwaves with non-accidental mortality, with corresponding distributions of CEHWI. a, East Asia; b, South Korea; c, Japan; d, China. The solid lines represent the estimated cumulative relative risks of mortality on days with specific heatwaves compared with days without heatwaves; shaded areas represent their 95% CIs. The vertical dotted lines from left to right indicate the 25th, 75th, and 90th percentiles of the regional cumulative excess temperature distribution of heatwaves, respectively.

Reviewer #3 (Remarks to the Author):

This study uses the Cumulative Excess Heatwave Index (CEHWI) to examine the associations between different types of heatwaves (daytime-only, nighttime-only, and compound) and mortality rates (non-accidental, cardiovascular, and respiratory) in 28 East Asian cities. The paper makes a valuable contribution by making the case for the use of continuous measures to study non-linear associations between heat and mortality. Consideration of different types of heatwaves across mortality causes is also a strength of the study. I want to offer the following suggestions for the authors' consideration.

Response: We sincerely appreciate your encouraging comments and constructive suggestions, which have greatly improved our manuscript. We have exerted our best to address your concerns. In particular, we revised the introduction to provide a more comprehensive context for the CEHWI and introduced the study hypothesis earlier. Additionally, we expanded the discussion to cover variability in adaptability thresholds across diseases, including a detailed explanation for the observed differences in daytime heatwave effects in China. A point-by-point response follows below.

Major Comments:

1. While the utility of the CEHWI is well-discussed in the discussion section, it would be beneficial to provide more context in the abstract and introduction. Perhaps the authors could elaborate on the CEHWI, addressing questions such as its definition, computation method, and advantages over existing measures in studying health impacts.

Response: Thank you for your valuable suggestions. We have carefully revised the manuscript to address your points regarding the CEHWI in both the introduction and abstract sections. Below is a summary of the changes we made:

1. Introduction

We have expanded the introduction to include more context about the CEHWI. Specifically, we now provide *a brief definition* of the CEHWI and highlight *its advantages* over traditional heatwave definitions. As per the structure typical of the journal, *detailed computational methods* are more appropriate for the Methods section (*Lines 403-410*). Nonetheless, the brief definition in the introduction now

includes a reference to the Methods for further details. **Please refer to lines 57-60 and 64-70 in the Introduction section or below:**

Lines 57-60: “However, unlike non-optimal temperature as a continuous variable²⁰⁻²³, prevailing risk assessment studies consistently oversimplified heatwaves as a binary variable⁸⁻¹⁴, which fails to capture these complex associations of heatwaves with mortality.”

Lines 64-70: “Recognizing the limitations of the prevailing binary definition, our study calculated the Cumulative Excess Heatwave Index (CEHWI), which measures cumulative excess heat during heatwaves relative to a predefined threshold (**detailed calculation described in Methods**). This index captures both the fluctuating magnitudes and continuous nature of cumulative excess heat during heatwaves, thus providing a more sophisticated lens on the complex and possible nonlinear relationships in the health effects of heatwaves.”

2. Abstract

In the abstract, we have briefly defined the CEHWI and mentioned its advantages. Due to the strict word limit, we have no choice but opted for a concise description of the index without very detailed descriptions of its computation method. Nonetheless, the current wording can still clearly and briefly indicate the definition and advantages of this index. **Please refer to lines 29-34 or below:**

Lines 29-34: “Heatwaves are commonly simplified as a binary variable in epidemiological studies, leaving heatwave-mortality exposure-response curves insufficiently established. This multi-country study across 28 East Asian cities employed the Cumulative Excess Heatwave Index (CEHWI) **representing the accumulation and persistence of excess heat**, to explore the potentially nonlinear associations of daytime-only, nighttime-only, and day-night compound heatwaves with mortality.”

2. The paper touches upon short-term adaptability when discussing non-linear trends in the findings. It might be helpful to introduce the concept of adaptability earlier in the paper, along with any hypothesized mechanisms the authors considered during the study.

Response: Thank you for this insightful suggestion. We have introduced the concept of short-term adaptability with hypothesized mechanisms earlier in the manuscript, as you suggested. **Please refer to lines 50-57 or below for details:**

Lines 50-57: “Evidence suggests that populations could exhibit adaptive physiological responses to elevated temperatures¹⁵⁻¹⁹, such as vasodilation, increased plasma volume, and higher sweat rates. These short-term, though bounded, physiological adaptations may lead to nonlinear responses to heatwaves with varying cumulative heat intensities or distinct heat exposure patterns. Given these adaptive capacities and the knowledge that the health effects of high temperature are not strictly linear in many prior studies²⁰⁻²³, we hypothesize that the relationship between heat anomalies during heatwaves and health outcomes is not necessarily linear. ”

3. The authors note that the "adaptability threshold varies across diseases and types of heatwaves" (line 218). While the discussion provides plausible reasons for variations across heatwaves, it might be worthwhile to expand on the variations observed across diseases. For instance, it would be interesting to explore why the exposure-response curve for respiratory diseases (in China and South Korea) differs significantly from all-cause mortality or cardiovascular diseases (in the case of daytime heatwaves). The authors' insights into the potential reasons for these variations across mortality causes would be valuable.

Response: Thank you for your insightful suggestion. In response, we have now expanded the discussion to address the variations observed across **diseases**. We structured the revision as follows: **first**, we describe the **overall trends** in risk differences across diseases at the East Asia level, followed by an explanation of the mechanisms behind these variations. This addresses why heatwave impacts on cardiovascular disease (CVD) and non-accidental mortality differ from those on respiratory disease (RD), as observed in South Korea. **Next**, we explore possible reasons for the distinct patterns between CVD and RD mortality in China. Below are the details:

1. General Trends in Sensitivity to Heatwaves across Diseases

We have added the following text to the manuscript:

Lines 244-262: “Similar to previous studies,^{9,13,30} our findings reveal differing risks and burdens associated with heatwaves across cardiovascular disease and respiratory disease mortality. Our estimated exposure-response curves further indicate that these differences may be due to the distinct response mechanisms of these two diseases to heatwaves. For cardiovascular disease, the exposure-response curves closely align with non-accidental mortality at low-to-medium CEHWI levels but become more pronounced at high CEHWI levels, although with regional differences. This may be due to the way heatwaves affect the cardiovascular system,

which is primarily caused by the failure of compensatory mechanisms^{37,38}. These failures typically occur at higher heat intensities and are marked by dehydration³⁹, electrolyte imbalances⁴⁰, and increased blood viscosity³⁸. In contrast, respiratory disease showed a rapid response to heatwaves, with significant impacts emerging at lower CEHWI levels, particularly for daytime or compound heatwaves. For example, in South Korea, the impact of daytime heatwaves on respiratory disease became significant at low CEHWI levels, whereas effects on non-accidental and cardiovascular disease mortality only surged after the CEHWI exceeded approximately the 90th percentile. This discrepancy may stem from the different physiological mechanisms involved. Respiratory conditions could be exacerbated due to increased respiratory rates and fluid loss⁴¹, which can occur at lower heat levels, leading to mucus thickening and airway inflammation⁴². These distinct physiological reactions emphasize the need for tailored disease-specific heatwave warning systems.”

2. Differences in Disease Responses in China

In the revised manuscript, we have also expanded the discussion on disease-specific heatwave responses in China, which did not exactly follow the overall trend described above. **Please refer to the text below:**

Lines 263-271: “The effects of daytime and compound heatwaves on respiratory disease in China showed an insignificant trend at low-to-moderate CEHWI levels, in contrast to cardiovascular disease, where effects emerged at similar CEHWI levels. This discrepancy may be due to the fact that individuals at high risk for respiratory disease might have already been severely affected by cardiovascular disease, which tends to respond to lower-intensity heatwaves in China. As a result, the impact of heatwaves on respiratory disease only became significant when heat intensities were high enough to affect those who could initially withstand lower-intensity heatwaves. Further research is needed to explore the physiological mechanisms underlying these observations.”

4. Regarding the previous point, the authors state that "Consistent with existing knowledge, our results affirm the heightened vulnerability of patients with cardiovascular and respiratory diseases to heatwaves, irrespective of heatwave types" (lines 263-265). However, this statement may not hold universally, as evidenced by the case of China for daytime heatwaves (Figures 1-3). It might be worth addressing this apparent inconsistency.

Response: Thank you for pointing out this inconsistency. We agree with your point

regarding the more heightened effects of daytime heatwaves on non-accidental mortality than on cardiovascular disease in China should be refined. Before detailing the potential reasons, we would like to first note that, in line with your previous suggestion, for easier cross-dimension comparisons, we have unified the lag structure to one week to estimate the exposure-response curves, rather than using “optimal” lag days for each type of heatwave based on their lag patterns. This unified approach has altered the shapes of the exposure-response curves in some regions and for certain diseases, particularly for daytime heatwaves.

Despite this change, we still observe that daytime heatwaves in China have a greater impact on non-accidental mortality compared to cardiovascular disease at higher CEHWI levels. This may be primarily due to:

(1) Our Chinese dataset included some of the most populous cities in East Asia, such as Shanghai, Beijing, and Guangzhou, given that our heterogeneity analysis indicated that regions with larger population sizes tend to experience more pronounced effects from daytime heatwaves on non-accidental mortality.

(2) Compared to South Korea and Japan, China’s relatively underdeveloped outdoor public infrastructure may exacerbate the impact of daytime heatwaves, as the population is likely exposed to prolonged periods of heatwaves without adequate cooling measures during the daytime.

(3) At higher CEHWI levels, daytime heatwaves also appeared to have a stronger impact on respiratory diseases than on cardiovascular diseases, contributing to the observed increase in non-accidental mortality, as respiratory diseases are included in the non-accidental category.

In response, we have **removed the original statement** about “*the heightened vulnerability of patients with cardiovascular and respiratory diseases to heatwaves, irrespective of heatwave types*”, and **revised it to**:

Lines 244-247: “*Similar to previous studies,^{9,13,30} our findings reveal differing risks and burdens associated with heatwaves across cardiovascular disease and respiratory disease mortality. Our estimated exposure-response curves further indicate that these differences may be due to the distinct response mechanisms of these two diseases to heatwaves.*”.

We have also added relevant explanations for why daytime heatwaves in China have a greater impact on non-accidental mortality:

Lines 276-284: “Cities with larger populations experience increased heatwave-related mortality risks, likely due to higher exposure in densely populated urban areas where the urban heat island effect is more pronounced^{43,45}. **This might also explain why effects of daytime heatwave in China exhibited longer lags and became significant for non-accidental mortality at lower CEHWI levels**, as our Chinese dataset included some of the most populous cities in East Asia, such as Shanghai, Beijing, and Guangzhou. Furthermore, the relatively underdeveloped public infrastructure in China may amplify the adverse impacts of daytime heatwaves.”

5. For the lag structures of all-cause mortality for daytime heatwaves in China (Fig S1), the authors note a longer lag. It would be beneficial to discuss potential reasons for this anomaly.

Response: Done as required. As mentioned in our previous response, the prolonged effects of daytime heatwaves (DHW) in China may be attributed to the inclusion of highly populous cities like Shanghai, Beijing, and Guangzhou, as population size is a significant factor in explaining DHW heterogeneity. Additionally, China's relatively underdeveloped outdoor public infrastructure, compared to South Korea and Japan, may further extend the duration of DHW effects. **Please refer to lines 276-284 in the revised manuscript or the text below:**

Lines 276-284: “Cities with larger populations experience increased heatwave-related mortality risks, likely due to higher exposure in densely populated urban areas where the urban heat island effect is more pronounced^{43,45}. **This might also explain why effects of daytime heatwave in China exhibited longer lags** and became significant for non-accidental mortality at lower CEHWI levels, as our Chinese dataset included some of the most populous cities in East Asia, such as Shanghai, Beijing, and Guangzhou. Furthermore, the relatively underdeveloped public infrastructure in China may amplify the adverse impacts of daytime heatwaves.”

6. Several anomalies are observed for China. Could this be due to the inclusion of the most northwestern city, which might have a significantly different climate? A sensitivity analysis excluding this city could provide valuable insights.

Response: Thank you very much for this insightful suggestion. As suggested, we conducted a sensitivity analysis excluding Urumqi, the most northwestern city in

China, and reanalyzed the results. However, we found no significant changes in the overall exposure-response curves for China, though the confidence intervals did widen slightly (*see Figure XI below*). This is likely due to Urumqi's relatively small sample size, indicating that its contribution to the overall pooled analysis was minimal.

Regarding the observed anomalies in China, we identified several key points that require further discussion:

➤ **1. Significant effects of Daytime heatwaves and non-accidental/CVD mortality at low-to-moderate CEHWI:**

In China, the impact of daytime heatwaves on non-accidental and cardiovascular disease mortality emerged even at low-to-moderate CEHWI levels (Figures 1 and 2). This could be attributed to the inclusion of highly populous cities of China, where the larger population sizes may amplify the effects of daytime heatwaves. Our heterogeneity analysis supports this observation, indicating that population size is a significant modifier of the effects of daytime heatwaves. Additionally, China's relatively underdeveloped outdoor public infrastructure, compared to more developed nations such as South Korea and Japan, might exacerbate the effects of daytime heat, leading to more pronounced health impacts (please refer to **lines 276-284**).

➤ **2. Prolonged lag effect of daytime heatwaves on mortality:**

We also observed that the effect of daytime heatwaves on mortality in China lasted longer compared to other regions. This prolonged effect could also be explained by the factors—larger population sizes and underdeveloped infrastructure—potentially leading to sustained exposure and delayed recovery during daytime heatwaves (please refer to **lines 276-284**).

➤ **3. Non-significant impact of Compound heatwaves on respiratory disease mortality at low-to-moderate CEHWI levels:**

In China, we found that compound heatwaves did not have a significant effect on respiratory disease mortality at low-to-moderate CEHWI levels. One plausible explanation is a competing risk: given that cardiovascular diseases in China show sensitivity to even low-intensity heatwaves, individuals at high risk for respiratory diseases may have already been severely affected or even succumbed to cardiovascular events before experiencing respiratory mortality. This may partially explain why the compound heatwave-related effects on respiratory diseases are less pronounced at these CEHWI levels (please refer to **lines**

263-270).

Lastly, we would like to emphasize that population adaptation to heatwaves is a multifaceted issue influenced by social, environmental, and physiological factors. More research is needed to explore the underlying mechanisms across different regions, diseases, and heatwave types. **We therefore added the following sentence in the revised manuscript:**

Lines 270-271: “Further research is needed to explore the physiological mechanisms underlying these observations.”

China

Daytime Heatwave

Nighttime Heatwave

Compound Heatwave

Figure X1. Pooled exposure-response relationship curves of the cumulative excess heatwave index (CEHWI) for daytime, nighttime, and day-night compound heatwaves with non-accidental mortality in China, **both with and without Urumqi included. The red curve is based on data excluding Urumqi, whereas the blue curve includes data from Urumqi.**

Note: Figure X1 was not included in the appendix to avoid an excessively long appendix.

7. In explaining the observed differences between daytime and nighttime heatwaves, the authors refer to differences in daytime vs. nighttime exposures. While this is a reasonable approach, it might be prudent to acknowledge the limitations of such inferences, given the lack of information about individuals' occupations in the sample. For instance, we cannot determine whether individuals primarily worked outdoors or indoors during the day.

Response: Thank you for your professional suggestion. We have now mentioned this limitation in the revised manuscript. **Please refer to the text below:**

Lines 224-236: “For the population's adaptability to DHW with low to medium CEHWI (below the 90th percentile), this could be interpreted by both short-term adaptation and long-standing acclimatization to the conventional hot-day and cool-night pattern^{19,33,34}. Similarly, NHW demonstrates significant harm only at medium-high CEHWI (above approximately 80th percentile), a phenomenon likely attributed to the nocturnal protection offered by building facilities, such as the implementation of cool and evaporative roofs¹⁵, and the use of air conditioning systems³⁵. **However, without information on individuals' daytime working environments (e.g., work indoors or outdoors), we acknowledge that the observed daytime heatwave adaptation may also be influenced by the effectiveness of buildings and air conditioning. Future studies should explore the health mechanisms behind daytime and nighttime heat exposure at the individual level to better understand the difference.**”

Minor Comments:

- Line 425: It would be helpful to specify the potential effect modifiers considered. The text mentions that details are provided in the appendix, but the specific variables are not indicated there either.

Response: Thank you for your kind reminder. The variables were originally described in the “*Second stage analysis*” section (lines 412-414 in the original manuscript), which was indeed not the most appropriate placement. Based on your suggestion, we have now moved the descriptions to the “*Effect modification analysis*” section for better clarity (See **lines 450-454 or the text below**). Additionally, we have included the sources of these variables in the “*Data Availability*” section to provide further transparency (see **lines 510-513 or the text below**).

- **Lines 450-454:** Considering the potential spatial heterogeneity, we employed univariable meta-regression models incorporating the following city-specific attributes separately: **Koppen-Geiger climate classification, gross domestic product (GDP) per capita, summer average temperature, summer temperature range, population size, latitude, and longitude.**
- **Lines 510-513:** City-specific attributes were obtained from Statistics Korea (<https://kostat.go.kr/ansk/>), the Statistics Bureau of Japan (<https://www.stat.go.jp/>), the National Bureau of Statistics of China (<https://www.stats.gov.cn/sj/ndsjs/>), and the Census and Statistics Department of Hong Kong (<https://www.censtatd.gov.hk/>).

- Line 437: Consider explaining or expanding on the abbreviations ECIs and BLUP for clarity.

Response: Done as required. Please refer to the text below:

Lines 464-469 (for BLUP): “we quantified the mortality burden (attributable deaths) due to CEHWI for each city based on their **best linear unbiased predictions (BLUPs)** of cumulative exposure-response curves^{49,59}. The BLUPs strike a balance between the city-specific curves derived from the first-stage analysis and the pooled results, enabling cities with shorter data series to borrow information from cities with larger populations and similar characteristics.”

Lines 145-147 (for the full term of eCIs): “Specifically, the AFs of CVD mortality related to DHW, NHW, and CHW were 0.14% [**95% empirical confidence intervals (eCIs):** 0.06%, 0.21%], 0.04% (0.01%, 0.05%), and 0.47% (0.36%, 0.52%), respectively.”. (We introduced the full term in the results section since it appears before the methods section).

Lines 469-472 (for calculations of eCIs): The AF was computed by dividing the number of attributable deaths by the total deaths in summer seasons. We subsequently derived 95% eCIs through **Monte Carlo simulations, based on the**

assumption of a multivariate normal distribution for the BLUP of the estimated coefficients.”

- Line 432: A brief description of the forward method would be beneficial for readers unfamiliar with the technique.

Response: Done as required. We have now removed this jargon, and directly described the method we used. Please refer to the text below:

Lines 462-463: “Utilizing a method that allows for more precise estimations of disease burden by considering the potentially intricate lag patterns of CEHWI-related mortality risks⁶¹, we quantified the mortality burden (attributable deaths) due to CEHWI for each city....”

I hope these comments prove helpful in further strengthening this already impressive study. The authors' work on the CEHWI and its application to different types of heatwaves provides valuable insights into the complex relationship between heat exposure and mortality.

Response: We sincerely appreciate your thoughtful and constructive comments, which have greatly contributed to strengthening our study. Particularly, your insights have helped us largely improve the logical flow of the introduction, expand our discussion to be more comprehensive and rigorous, and make the methods section clearer and more accessible. We are truly grateful for your input in enhancing the overall quality of our work.

REVIEWERS' COMMENTS

Reviewer #1 (Remarks to the Author):

Your revisions address my concerns and, from my vantage point, your manuscript appears suitable for publication.

Reviewer #3 (Remarks to the Author):

Upon review of the authors' rebuttal and revised manuscript, I commend the diligence and thoroughness with which they have addressed all comments. The integration of these comments into the manuscript is comprehensive and well-executed, rendering great clarity and coherence in the revised manuscript.

Notably, the elaborations in the introduction and discussion section enhance the motivation of the study and provide valuable context for the study's findings. The authors' commitment to refining their work is evident throughout the revised manuscript and their rebuttal.

Reviewer #1 (Remarks to the Author):

Your revisions address my concerns and, from my vantage point, your manuscript appears suitable for publication.

Response: We sincerely thank you for your thoughtful and constructive feedback throughout the review process. Your valuable suggestions have greatly contributed to improving the quality and clarity of our manuscript.

Reviewer #3 (Remarks to the Author):

Upon review of the authors' rebuttal and revised manuscript, I commend the diligence and thoroughness with which they have addressed all comments. The integration of these comments into the manuscript is comprehensive and well-executed, rendering great clarity and coherence in the revised manuscript.

Notably, the elaborations in the introduction and discussion section enhance the motivation of the study and provide valuable context for the study's findings. The authors' commitment to refining their work is evident throughout the revised manuscript and their rebuttal.

Response: Thank you very much for your time and efforts in improving our paper. We greatly appreciate your constructive comments, especially your suggestions on clarifying the logic in the introduction and addressing disease and regional heterogeneity in the discussion. These insights have greatly enhanced the clarity and quality of our manuscript.